# In-depth investigation of genome to refine QTL positions for spontaneous sex-reversal in XX rainbow trout

**Audrey Dehaullon[1†], Clémence Fraslin[1,2†], Anastasia Bestin[3], Charles Poncet[4], Yann Guiguen[5], Edwige Quillet[1], Florence Phocas** [1]*

1 Université Paris-Saclay, INRAE, AgroParisTech, GABI, 78350, Jouy-en-Josas, France, 2 The Roslin Institute and Royal (Dick) School of Veterinary Studies, The University of Edinburgh, Midlothian, United Kingdom, 3 SYSAAF, Station INRAE-LPGP, Campus de Beaulieu, Rennes cedex, France, 4 Université Clermont-Auvergne, INRAE, GDEC, 63039 Clermont-Ferrand, France, 5 INRAE, LPGP, Campus de Beaulieu, Rennes cedex, France

† Co-first authors: Audrey Dehaullon and Clémence Fraslin.

* florence.phocas@inrae.fr

## Abstract

Sex determination is a flexible process in fish, controlled by genetics or environmental factors or a combination of both depending on the species. Revealing the underlying molecular mechanisms may have important implications for research on reproductive development in vertebrates, as well as sex-ratio control and selective breeding in fish. Phenotypic sex in rainbow trout (*Oncorhynchus mykiss*) is primarily controlled by a XX/XY male heterogametic sex determination system. Unexpectedly in genetically XX all-female farmed populations, a small proportion of males or intersex individuals are regularly observed. Spontaneous masculinisation is a highly heritable trait, controlled by minor sex-modifier genes that remain unknown, although several Quantitative Trait Loci (QTL) were detected in previous studies. In the current work we used genome-based approaches and various statistical methods to further investigate these QTL. DNA markers that were previously identified in a French commercial population on chromosomes Omy1, Omy12 and Omy20 were validated in six different farmed trout populations. Functional candidate genes that may be involved in spontaneous masculinisation by reducing germ cell proliferation and repressing oogenesis of XX-rainbow trout in the absence of the master sex determining gene were identified. In particular, *syndig1*, *tlx1* and *hells* on Omy1, as well as *khdrbs2* and *csmd1* on Omy20 deserve further investigation to validate their potential sex-modifier roles as well as their interaction with rearing temperature. Those findings could be used to produce all-female populations that are preferred by farmers due to a delayed maturation of females and higher susceptibility of male trout to diseases.

**Data availability statement:** Raw sequence data that were used in this study are deposited in the ENA (Project PRJEB75960 in https://www.ebi.ac.uk/ena/browser/view/PRJEB75960).

**Funding:** The European Maritime and Fisheries Fund and the French National Government supported the NeoBio project (R FEA470016FA1000008). The funders had no role in study design, data collection and analysis, decision to publish, or preparation of the manuscript.

**Competing interests:** The authors have declared that no competing interests exist.

## Introduction

Sex reversal has been defined as a mismatch between the phenotypic and the genetic sex of an individual [1]. In fish, sex reversal can be induced by hormonal treatments (estrogens to feminize and androgens to masculinize) during critical fry stages [2,3]. Manipulating the sexual phenotype of fish has been used to prevent early maturity and reproduction, as well as increase production in various cultured species or create subsequent sterile generations for fisheries management [3]. Sex-reversed individuals can be used to produce populations that are genetically all-male or all-female (also known as 'monosex'). Rainbow trout (*Oncorhynchus mykiss*) is a worldwide cultured salmonid species. The production of large rainbow trout over 1 kg body weight has become a growing commercial interest with the development of a market for fresh and smoked fillets. A major bottleneck for this commercial market is the early maturation of males (1–2 year) relative to females (2–3 years) that causes stopped growth, increased sensitivity to pathogens (*Saprolegnia* spp.), and reduced flesh quality [4,5,6]. Therefore, all-female rainbow trout stocks are often preferred in aquaculture. These stocks are currently produced by crossing hormonal sex-reversed genetically XX-females, known as neomales (XX males), with XX-females [2]. The growing concerns or potential risks to human and environmental health make the prevalent approach of producing neomales via the oral application of the hormone 17-alpha methyltestosterone less sustainable. The finding of a sex control alternative to hormonal treatment based on a safe, consumer and environmentally friendly method is a major challenge for the production of all-female populations [7].

Sex determination and sex differentiation mechanisms in fish are diverse and complex. Most of the teleosts are gonochorists meaning that all individuals within a species develop either as males or females and remain the same throughout their lives. Various systems of sex determination have been observed in gonochoristic fishes [1], spanning from a genetic sex determination (GSD) system with a genetic monofactorial system, XX/XY or ZZ/ZW, to different multifactorial systems, or environmental sex determination (ESD). During sex differentiation, the testis and ovary pathways are mutually antagonistic and compete for control of gonad fate. Understanding sex-differentiation processes requires to unravel the origin and developmental pathways of cells and organs involved in the formation of the primordial gonad. During primary sex differentiation in vertebrates, the primordial bipotential gonad commits to either an ovary or a testis developmental fate. Within both ovary and testis, a clear distinction between germ and somatic cells can be made, with the former having the potential to mitotically divide and enter meiosis, and the latter differentiating into associated structural and endocrine cells. In teleosts, primordial germ cells (PGCs) are established and specialized at the early blastula stage and migrate to the genital ridge during embryonic development [8].

Spontaneous sex reversal occurs because of a failure to maintain the initiated pathway or a failure to repress the opposite pathway [9], switching the sexual phenotype of the organism to the opposite sex. Once the gonad has

committed to a particular fate, it begins to produce sex-specific hormones that will have local and direct effects on germ cell development, but will also drive the secondary process of sex differentiation in which the somatic tissues of the organism differentiate toward one sex or the other. The developmental pathway leading to steroid production in gonadal somatic cells requires complex regulation of multiple genes [10] involved in the differentiation of steroid-producing cells.

Over the last two decades evidence has accumulated that both somatic and germ cells play critical roles in fish gonadal differentiation. However, the exact mechanisms and factors involved remain to be elucidated [11,12]. As evidenced from studies in mammals [13], somatic cells may first differentiate in response to the activation of master sex-determining genes, and subsequent differentiation of PGCs into male or female gametes follows in the differentiating testis or ovary in response to signals derived from surrounding somatic cells. Alternatively, it is possible that, first, PGCs interpret internal genetic or external environmental cues and directly transform into spermatogonia or oogonia; then, the surrounding somatic cells may be induced by the germ cells to differentiate accordingly to provide an appropriate hormonal environment for further gonadal differentiation [14]. It is thought that control of estradiol synthesis could play a key role not only for ovarian, but also for testicular differentiation and sex change in fish [15]. This working hypothesis states that the gonadal aromatase gene, *cyp19a1a*, up-regulation would be needed not only for triggering, but also for maintaining ovarian differentiation; and that *cyp19a1a* down-regulation would be the only necessary step for inducing a testicular differentiation pathway. In Atlantic salmon as well as in loach and goldfish, it was shown that germ cells are not essential for gonadal sex differentiation [16]. Germ cell-free female gonads develop an ovarian somatic structure in these species, unlike in zebrafish or medaka where all germ cell-free fish develop somatic testes. Evidence from both zebrafish and medaka models [17,18,19,20] suggests important feminizing roles of the germ cells in gonadal differentiation and thus corroborates the alternative hypothesis. In particular, when germ cells are ablated in medaka, XX fish show female-to-male sex reversal, while XY fish exhibit male-to-female sex reversal [19]. Any disruption of this germ cell/somatic cell cross-talk would then disrupt ovarian somatic cell differentiation, leading to an absence of estrogen synthesis and a subsequent masculinization. More subtle regulations not involving the complete loss of germ cells, but instead some differential germ cell proliferation rates, could be also suggested as important triggers of gonadal sex differentiation. For example, the Japanese medaka sex determining gene, *dmrt1Y*, has been found to be an inhibitor of germ cell proliferation [21] and is only expressed during testicular differentiation [22].

In GSD species, the master sex determining trigger is encoded by a gene on a sex chromosome, which activates a network of downstream regulators of sex differentiation. Currently, there have been more than 20 sex determining genes discovered in fish, mammals, birds and frogs, and as many as 13 are derived from the transforming growth factor Beta (TGF-β) pathway [23]. This important pathway in sex determination/differentiation of fishes potentially regulate the number of germ cells, and/or inhibit aromatase activity to determine and maintain sex (e.g., Chen et al. [24]). Interestingly, *sdY* (sexually dimorphic on the Y-chromosome gene), the master sex-determining gene of rainbow trout and other salmonid species, does not belong to the usual master sex gene families (the DM domain and Sox protein families), nor does it belong to TGF-β signaling pathway. It is a truncated duplicated copy of the immune response gene *irf9* (Interferon Regulatory Factor 9) [25]. While very specific to salmonids, the gene *sdY* is acting directly on the classical sex differentiation regulation network in all vertebrates as its protein has been shown to interact with *foxl2* (Forkhead Box L2*)* [26] that is a well-known conserved female differentiation factor [27]. *foxl2* was shown to be involved in the goat polled intersex syndrome, a deletion of the gene triggering early testis differentiation and XX female-to-male sex reversal [28,29].

Although phenotypic sex in rainbow trout is primarily determined by a male heterogametic XX/XY GSD system, spontaneous masculinization of XX fish is a phenomenon that has been repeatedly observed in various rainbow trout populations, at generally limited frequencies (~1–2%) with some individuals being only partially affected (intersex) [30]. Spontaneous sex reversal is a highly heritable trait [30] and several Quantitative Trait Loci (QTL) associated with this

trait were detected [27,31], highlighting the existence of several minor sex-determining genes that are independent of the major sex determinant carried by the sex chromosome. In particular, in a previous study [30], we identified in a French trout population two QTL located on chromosome 1 with a strong evidence while two suggestive QTL were identified on chromosomes 12 and 20 using the rainbow trout reference genome of Swanson line [32]. All SNP variants from these four QTL regions that covered tens of genes were further investigated in the present study.

Firstly, we sought to validate the existence of the four QTL in six different French rainbow trout populations, and secondly, we wanted to refine their location in the initial discovery population by combining machine learning and principal components approaches, and considering the new reference genome derived for Arlee line [33].

## Materials and methods

### Sequenced samples from the discovery population

The first data set was initially produced by Fraslin et al. [30]. The 4 QTL regions that were discovered were further studied in depth in this work (Fig. 1).

The fish samples and records came from the French rainbow trout farm "Les Fils de Charles Murgat" (Beaufort, France; UE approval number FR 38 032 001). The 60 dams sequenced genomes were initially aligned against the reference assembly genome Omyk_1.0 (GenBank assembly accession GCA_002163495.1) of the double haploid line Swanson from Washington State University [32]. Following the same procedure as described in Fraslin et al. [30], these 60 sequences were realigned against the reference genome USDA_OmykA_1.1 of the WSU double haploid line Arlee (GenBank assembly accession GCA_013265735.3) [33]. This realignment was performed because French rainbow trout populations were known to be phylogenetically closer to Arlee's genome that to Swanson's one [34]. On average, 9.252E+7 paired reads (1.456E+7 sd) were mapped on the Arlee genome, corresponding to a mean coverage of 12X. While the percentage of properly paired sequence reads over all sequenced pairs was in average 87.3% for the alignment against Swanson's genome assembly, these statistics went up to 94.1% for the alignment against Arlee's genome assembly.

After mapping, duplicates were marked using the MarkDuplicate tool, and base quality score where recalibrated using ApplyBQSR tool from GATK 4.2.2.0 [35], then variants were called independently with the three different variants calling tools HaplotypeCaller from GATK, Freebayes 1.3.5 [36] and SAMtools Mpileup 1.11 [37]. Recommended quality filters were used to ignore low-confidence alignments: a minimum mapping quality of 30, a minimum base quality required to consider a base for calling of 10X reads, minimum phred-scaled confidence threshold at which variants should be called of 30. A total of 29,229,949 variants that were obtained by the three different variant calling tools were kept for further quality controls using vcftools 1.15 and keeping indels and SNPs located only on known chromosomes, removing variants located on un-located contigs or mitochondrial chromosome. Further quality filtering on variant coverage were performed using SelectVariant and Variant Filtration tools (GATK) according to the hard-filtering recommendations [38]. Briefly $QD < 2.0$ (to normalize variant quality), $FS > 60.0$ (measure of the Phred-scaled probability that there is strand bias at the site), $MQ < 40.0$ (mapping quality), $SOR > 3.0$ (Strand Odds Ratio), $MQRankSum < -12.5$ (u-based z-approximation from the Rank Sum Test for mapping qualities) and $ReadPosRankSum < -8.0$ (u-based z-approximation from the Rank Sum Test for site position within reads). Final numbers of 21,517,540 SNPs and 4,850,647 indels, for a total of 26,368,187 variants, were kept for further analysis.

Among the 60 sequenced females, a subset of 23 dams with at least 10 progeny with phenotypic sex recorded and extreme proportions of sex-reversed offspring, either at least 25% or below 6% of neomales among their XX daughters (see Table A in S1 Tables) were used in the current study. For each of the 3 chromosomes investigated in the present study, all SNPs that exhibited polymorphism among the genomes of those 23 dams were kept for the final analysis, i.e., 854,157 SNPs for Omy1, 965,320 SNPs for Omy12 and 492,959 SNPs for Omy20.

## French rainbow trout populations under study

**Discovery population**
**To refine mapping** of the four QTL identified in Fraslin et al. [30] using a new genome assembly

**Validation populations**
**To confirm the existence** of the four QTL detected in [30]

French commercial population:
'Les Fils de Charles Murgat'
23 dam sequence genomes with extreme progeny phenotypes

Six rainbow trout populations, each with 30 to 77 sex phenotyped fish:
A: related cohort from 'Les Fils de Charles Murgat' discovery population
B, C: from French breeding company 1
D: Unknown origin
E, F: from French Breeding company 2

## Applied Methods

**Fisher's exact test** on all SNPs in extended genomic regions
**To define new boundaries for the QTL regions**

**Fisher's exact test** on 173 SNPs
**To validate the existence in various populations of the four QTL regions**

**Random Forests on haplotypes** (including 90,144 SNPs) within the newly defined QTL regions
To identify the most relevant haplotypes and positional candidate genes (n=45)

**Discriminant Analysis of Principal Components** on 27,828 SNPs (based on the 45 genes previously identified)
**To find the variants that best discriminate between dams with HIGH or LOW masculinised progeny**

**Fig. 1. Graphical summary of the different statistical analyses performed in the discovery or validation populations.**

### Genotyped samples from six validation populations

We sought to validate the existence of the four QTL detected by Fraslin et al. [30] as linked to spontaneous sex-reversal of XX trout in six diverse French populations of rainbow trout (Table 1, Fig. 1). Population A was composed of XX sibs from the same birth cohort of the parents used to produce the initial QTL discovery population [30]. Fish from populations

**Table 1. Description of the 6 rainbow trout XX-populations used for QTL validation.**

| Population | A | B | C | D | E | F |
|---|---|---|---|---|---|---|
| Number of genotyped fish | 49 | 30 | 64 | 77 | 45 | 50 |
| Phenotyped as female | 34 | 22 | 32 | 29 | 31 | 32 |
| Phenotyped as male (including intersex) | 15 | 8 | 32 | 48 | 14 | 18 |
| Phenotyped as pure male (no intersex) | 8 | nr | 17 | 32 | 6 | nr |

nr: values not reported by the breeding company

B and C, and those from populations E and F came from two other French breeding compagnies, while fish from population D came from a commercial site using fry from unknown origin.

Two 96 SNPs genotyping arrays using microfluidic real-time PCR Fluidigm Kasp chemistry were designed. A set of 192 SNPs identified in Fraslin et al [30] (see Table B in S1 Tables) were selected, corresponding to: 140 SNPs in the two main QTL identified on chromosome Omy1, 19 SNPs in a putative QTL region detected on Omy12 and 33 SNPs in a large 6 Mb region with putative QTL on Omy20.

Pieces of caudal fin sampled from 315 fish, corresponding to at least 30 XX fish per population, including a minimum of eight neomales (Table 1), were sent to Gentyane genotyping platform (INRAE, Clermont-Ferrand, France) for genotyping after DNA extraction using the DNA advance kit from Beckman Coulter following manufacturer instructions. After quality control, 19 SNPs were eliminated (including 14 from Omy1) from the analyzes of all the populations because their genotyping rate was below 90%, 173 SNPs were thus kept for QTL validation.

## Statistical analysis for QTL confirmation in six validation populations

An exact Fisher test was run for each of the six validation populations using the fisher.test function in R version 4.3.1 [39] which implements the method developed by Mehta and Patel [40,41] and improved by Clarkson, Fan and Joe [42] to test the independence of genotypes at each SNP and phenotypic sex of fish.

This test was applied for the set of 173 SNPs across all six populations (Fig. 1). In any of these populations, ~170 SNPs were polymorph and had a genotyping rate above 90%. To account for multiple testing using Bonferroni correction, SNPs were considered significant at the genome level in a given population when $p\_value < 0.0005$ (i.e., $\log_{10}(p\_value) < 3.3$, assuming that about 100 SNPs were not highly redundant among ~170 tested for each population). SNPs only had a putative effect when $p\_value < 0.005$ (i.e., $\log_{10}(p\_value) < 2.3$).

## Exact fisher test on all SNPs of QTL in 23 extreme dams of the discovery population

In order to refine the QTLs boundaries and confront the results between the validation and discovery populations (Fig. 1), an exact Fisher test was run to detect the association of all the sequence variants observed in the four extended QTL regions with the average progeny sex-ratios of the 23 extreme dams of the discovery population.

A first putative threshold of $p\_value < 0.005$ (i.e., $\log_{10}(p\_value) < 2.3$) was used to identify putative QTL. A second, more stringent, threshold ($p\_value < 0.002$) was considered to focus on the main significant associations with sequence variants in the discovery population. Because of the strong linkage disequilibrium observed across SNPs along the chromosome [43], the QTL region on Omy20 was expanded to test, one-by-one, all the variants spanning the chromosome from 27 to 38 Mb.

## Statistical approaches for refinement of QTL locations in the discovery population

To refine the QTL locations in the discovery population using the latest genome assembly we i) used Random Forests approaches with haplotypes and ii) applied a Discriminant Analysis of Principal Components on the sequence data from the 23 dams selected above (Fig. 1.).

## Machine learning approach using random forests

Random forest (RF) is a machine learning method that aggregates complementary information from an ensemble of classification or regression trees trained on different bootstrap samples (animals) drawn with replacement from the original data set [44]. First, every tree is built using a bootstrap sample of the observations. Second, at each node, a random subset of all predictors (the size of which is referred to as mtry hereafter) is chosen to determine the best split rather than the full set. Therefore, all trees in a forest are different. A particularity of the RF is the out-of-bag data, which corresponds to the animals not included (roughly 1/3) in the bootstrap sampling for building a specific tree. It can be used as an internal validation set for each tree, which allows the computation of RF error rate based on misclassified animals in the out-of-bag data. RF can be applied successfully to "large p, small n" problems, and it is also able to capture correlation as well as interactions among independent variables [45].

Five genomic zones were considered to define haplotypes of subsequent SNPs spanning all the QTL regions from 67 to 69Mb on Omy1, from 8.5 to 9.5Mb on Omy12, and from27 to 29Mb, 33.5 to 35.5Mb, and 36–37Mb on Omy20. In total 21,310 variants for Omy1, 12,191 for Omy12, and 56,643 for Omy20, were sonsidered in the RF analysis.

The genotypes were recorded depending of the number of reference alleles present in the genotype: "2" for a homozygous reference genotype, "1" for a heterozygous genotype, "0" for the alternative homozygote; missing genotypes were encoded "5".

To minimize the RF error rates, a preliminary set of analyses was run to establish the optimum SNP window size (swind), the number of sampled trees (ntree), and the number of sampled predictor variables (mtry) chosen at random to split each node. In any analysis, when variables are highly correlated, they can act as substitutes for one another. This weakens the evidence of association for any single correlated variable with the outcome if all variables are included in the same model.Therefore, to avoid redundancy of information and limit the number of variables to analyze, haplotypic genotypes were built by retaining one SNP out of two and gathering them in the same haplotype across a window of 80 initial SNPs (correspond to swind=40). We varied swind from 20 to 200 subsequent SNPs, ntree from 100 to 1,000 and mtry from 5 to 20. The final parameter set (swind=40, ntree=400 and mtry=7) was chosen as consistently giving the lower median error rate over 50 runs for any of the three tested chromosomes. A final set of 246 haplotypes for Omy1 (131 for Omy1_a and 115 for Omy1_b), 152 haplotypes for Omy12 and 707 haplotypes for Omy20 (247, 288, and 172, respectively for Omy20_a, Omy20_b and Omy20_c) were then analyzed by averaging results across 100 RF runs. We used the package RandomForest 4.18 [46] in R version 4.3.1 [39].

Haplotypes were ranked by decreasing order of importance in the prediction model, using the mean decrease in Gini index. The Gini index measures the probability for a specific feature of being misclassified when chosen randomly. The higher is the value of mean decrease in Gini index, the higher is the importance of the variable in the model.

A list of 45 positional candidate genes was established based on results of the top-ranked haplotypes as well as the preliminary Fisher exact tests on SNP.

## Discriminant analysis of principal components to identify the most promising variants

The Discriminant Analysis of Principal Components (DAPC) helps explore the genetic structure of biological populations. This multivariate method consists in a two-step procedure: 1) genetic data are centred and used in a Principal Component Analysis (PCA); 2) a Linear Discriminant Analysis is applied to the principal components of PCA. A basic matrix operation is used to express discriminant functions as linear combination of alleles, allowing one to compute allele contributions, i.e., to identify the most interesting variants [47].

The 33,731 SNPs that were in the list of genes identified by RF or Fisher's exact test as well at the SNPs in the close upstream and downstream regions of theses genes (±2 kb) were used in the DACP. Filtering out SNPs with call rate below 100% using PLINK 1.9 [48], DAPC was applied on a final set of 27,828. The adegenet R package [49] was used to identify, by DAPC, the variants that best discriminate between dams with high and dams with low sex-reversal ratios in their offspring.

### Annotation of genes and variants

Genes within QTL regions were annotated using the NCBI *O. mykiss* Arlee genome assembly USDA_OmykA_1.1. (GCA_013265735.3) (Gao et al. 2021). In addition to NCBI gene summaries, functional information for genes was extracted from the human gene database GeneCards® (https://www.genecards.org/) that also includes protein summaries from UniProtKB/Swiss-Prot (https://www.uniprot.org/uniprotkb/).

SNP annotation was performed for the 33,731 SNPs identified for the list of genes of interest, applying the software SNPEff4.3T [50] on the NCBI annotation file annotation released for *O. mykiss* Arlee genome reference assembly (GCA_013265735.3). These annotations permitted to retrieve any information about the predicted impact of the variants on the genes, and, in some cases, on the resulting proteins. Relevant variants were further studied using the Genome Data Viewer of NCBI to indicate their intronic or exonic positions within the genes.

### Ethics statement

All applicable international and national guidelines for sampling, care, and experimental use of organisms for the study have been followed. As part of the standard breeding practices for a commercial breeding program, the handling of fish was not subject to oversight by an institutional ethic committee. Animals were treated in compliance with the European Communities Council Directive (9858/CEC) for the care and use of farm animals.

## Results

### Validation of the existence of the four QTL in various French rainbow trout populations

Among the 173 SNPs that could be tested for their association with spontaneous sex-reversal in the validation populations, 117 had a significant effect in at least one population with 50 SNPs on Omy1 having an effect in population A, that was genetically close to the QTL discovery population. Most of these 50 SNPs had also an effect observed in populations B, C and E. However, none of the SNPs tested on Omy12 or Omy20 had an effect in population A, probably because of sampling issues for low frequency variants in the population but 7 SNPs on Omy12 (out of 17 tested) had an effect in populations C, D, E and F; and 13 SNPs on Omy20 (out of 30 tested) had an effect in populations C, D and F.

The list of the 90 SNPs that had at least a putative associated effect in two validations populations is given in Table C in S1 Tables. The 63 SNPs that had at least a putative effect in four validation populations, or a clear significant effect in two or more populations is given in Table 2 along with the NCBI annotation of 50 SNPs located within or in the close vicinity (< 5kb) of genes.

### Extension of the QTL region boundaries in the discovery population

Because most of the SNPs located at the boundaries of the QTL regions reported by Fraslin et al. [30] had a significant effect in some of the validation populations, a Fisher's exact test was applied on the dams' average progeny phenotypes in the discovery population for all sequence variants located in QTL regions (enlarged by 0.5 Mb at least on each side). Based on these tests, new extended boundaries for the QTL were defined (see Table 3). In particular, QTL in three different regions of Omy20 were enlarged, spanning from 27 to 29 Mb, 33.5 to 35.5 Mb, and 36–38 Mb, respectively for Omy20_a, Omy20_b and Omy20_c.

In total, 246 SNPs had a p_value ≤ 0.005 (see Table D in S1 Tables), and the 19 SNPs with a more stringent p_value ≤ 0.002 were mainly located within genes or in their close vicinity (Table 4).

### Identification of the most relevant haplotypes and genes explaining sex-reversal in the discovery population

Initially, a Random Forest (RF) analysis was performed for each of the three chromosomes with all their QTL regions jointly analyzed. Respectively for the RF on Omy1, Omy12 and Omy20, the 30, 15 and 30 top- ranked haplotypes based

**Table 2. Most significant SNPs in two or more out of the six validation populations named A to F (names in bold correspond to a p_value < 0.0005, others to a p_value < 0.005).**

| Omy | Position | P1 | P2 | P3 | P4 | NCBI gene annotation (Genecards symbol) |
|---|---|---|---|---|---|---|
| 1 | 67186650 | **A** | B | **C** | F | synapse differentiation-inducing gene protein 1 (*syndig1*) |
| 1 | 67187724 | **A** | B | **C** | F | synapse differentiation-inducing gene protein 1 (*syndig1*) |
| 1 | 67273221 | **A** | B | **C** | F | 1.982 kb upstream acyl-CoA synthetase short chain family member 1 (*acss1*) |
| 1 | 67319795 | **A** | B | **C** | F | 28.57 kb upstream visual homeobox 1 (*vsx1*) |
| 1 | 67340956 | **A** | B | **C** | F | 15.11 kb upstream ectonucleoside triphosphate diphosphohydrolase 6-like (*entpd6*) |
| 1 | 67353129 | A | **B** | **C** | **F** | 2.941 kb upstream *entpd6* |
| 1 | 67387599 | **B** | **C** | **F** | | barrier-to-autointegration factor (*banf1*) |
| 1 | 67387902 | **B** | **C** | **F** | | barrier-to-autointegration factor (*banf1*) |
| 1 | 67389450 | **B** | **C** | **F** | | phosphorylase, glycogen; brain (*pygb*) |
| 1 | 67395895 | A | **B** | **C** | F | phosphorylase, glycogen; brain (*pygb*) |
| 1 | 67406669 | B | **C** | | | phosphorylase, glycogen; brain (*pygb*) |
| 1 | 67410770 | B | **C** | | | phosphorylase, glycogen; brain (*pygb*) |
| 1 | 67413276 | B | **C** | D | | phosphorylase, glycogen; brain (*pygb*) |
| 1 | 67434621 | A | **B** | **C** | | abhydrolase domain containing 12, lysophospholipase (*abhd12*) |
| 1 | 67435714 | **B** | **C** | | | abhydrolase domain containing 12, lysophospholipase (*abhd12*) |
| 1 | 67461050 | **B** | **C** | | | ninein-like (*ninl*) |
| 1 | 67464776 | **A** | **B** | **C** | F | ninein- like (*ninl*) |
| 1 | 67494288 | **B** | **C** | | | ninein- like (*ninl*) |
| 1 | 67522146 | **B** | **C** | | | 6.465 kb upstream pleckstrin and Sec7 domain containing a (*psda*) |
| 1 | 67535219 | **B** | **C** | | | pleckstrin and Sec7 domain containing a (*psda*) |
| 1 | 67546988 | **B** | **C** | | | pleckstrin and Sec7 domain containing a (*psda*) |
| 1 | 67584462 | **B** | **C** | | | shieldin complex subunit 2 (*shld2*) |
| 1 | 67626764 | **B** | **C** | | | glutamate dehydrogenase 1 (*glud1*) |
| 1 | 68248684 | A | B | **C** | E | 26.69 kb upstream solute carrier family 2 member 15a (*slc2a15*) |
| 1 | 68270478 | A | B | **C** | E | 4.901 kb upstream of solute carrier family 2 member 15a (*slc2a15*) |
| 1 | 68305255 | **A** | B | **C** | | 9.849 downstream solute carrier family 2 member 15a (*slc2a15*) |
| 1 | 68310800 | **A** | B | **C** | | : 5.532 kb upstream fibroblast growth factor 8a (*fgf8a*) |
| 1 | 68312110 | **A** | B | **C** | | 4.222 kb upstream of fibroblast growth factor 8a (*fgf8a*) |
| 1 | 68315508 | A | B | **C** | | fibroblast growth factor 8a (*fgf8a*) |
| 1 | 68316754 | **A** | B | **C** | | fibroblast growth factor 8a (*fgf8a*) |
| 1 | 68329760 | A | **C** | | | 6.823 kb downstream fibroblast growth factor 8a (*fgf8a*) |
| 1 | 68330279 | A | **B** | **C** | | 8.758 kb upstream F-box and WD repeat domain containing 4 (*fbxw4*) |
| 1 | 68339634 | **A** | **C** | | | F-box and WD repeat domain containing 4 (*fbxw4*) |
| 1 | 68466812 | **C** | **F** | | | 0.138 kb downstream of helicase, lymphoid specific (*hells*) |
| 1 | 68467934 | **A** | B | **C** | | helicase, lymphoid specific (*hells*) |
| 1 | 68470380 | **A** | B | **C** | | helicase, lymphoid specific (*hells*) |
| 1 | 68495916 | **A** | B | **C** | | 0.826 kb downstream of uncharacterized LOC118965305 |
| 1 | 68496653 | **A** | B | **C** | | 1.563 kb downstream of uncharacterized LOC118965305 |
| 1 | 68501373 | **A** | B | **C** | | 6.283 kb downstream uncharacterized LOC118965305 |
| 1 | 68503159 | **A** | B | **C** | | 8.069 kb downstream uncharacterized LOC118965305 |
| 1 | 68504970 | A | B | **C** | | 9.880 kb downstream uncharacterized LOC118965305 |
| 1 | 68505133 | **A** | B | **C** | | 10.04 kb downstream uncharacterized LOC118965305 |
| 1 | 68505220 | **A** | B | **C** | | 10.13 kb downstream uncharacterized LOC118965305 |
| 1 | 68505500 | **A** | B | **C** | | 10.41 kb downstream uncharacterized LOC118965305 |
| 1 | 68509786 | **A** | B | **C** | | 10.01 kb upstream LOC110527930 |

*(Continued)*

**Table 2.** (Continued)

| Omy | Position | P1 | P2 | P3 | P4 | NCBI gene annotation (Genecards symbol) |
|---|---|---|---|---|---|---|
| 1 | 68511322 | **A** | B | **C** | | 8.472 kb upstream LOC110527930 |
| 1 | 68519589 | **A** | B | **C** | | 0.205 kb upstream of LOC110527930 |
| 1 | 68521100 | **A** | B | **C** | | uncharacterized LOC110527930 |
| 1 | 68522155 | **A** | B | **C** | | uncharacterized LOC110527930 |
| 1 | 68524318 | **A** | B | **C** | | uncharacterized LOC110527930 |
| 1 | 68529520 | **A** | B | **C** | E | uncharacterized LOC110527930 |
| 1 | 68529535 | **A** | B | **C** | E | uncharacterized LOC110527930 |
| 1 | 68529584 | **A** | B | **C** | E | uncharacterized LOC110527930 |
| 1 | 68529621 | **A** | B | **C** | E | uncharacterized LOC110527930 |
| 1 | 68529794 | **A** | B | **C** | E | uncharacterized LOC110527930 |
| 1 | 68538278 | **A** | **B** | **C** | | uncharacterized LOC110527930 |
| 1 | 68546628 | **A** | **B** | **C** | | uncharacterized LOC110527930 |
| 1 | 68605451 | **A** | **B** | **C** | | collagen alpha-1(XIII) chain (*col13a1*) |
| 12 | 9164517 | **D** | F | | | hyperpolarization-activated cyclic nucleotide-gated cation channel 1 (*hcn1*) |
| 20 | 34350986 | **D** | **F** | | | 3-hydroxymethyl-3-methylglutaryl-CoA lyase-like 1 (*hmgcll1*) |
| 20 | 34353785 | **D** | **F** | | | 3-hydroxymethyl-3-methylglutaryl-CoA lyase-like 1(*hmgcll1*) |
| 20 | 37496331 | C | **F** | | | ENSOMYG00000068776 |
| 20 | 37497790 | **D** | F | | | ENSOMYG00000068776 |

on Gini index as well as the corresponding genes are reported in Tables E, F and G in S1 Tables. For the 2-Mb region on Omy1 (246 haplotypes spanning from 67 to 69 Mb), the median error rate was 28% with a minimum error rate of 15% and a maximum rate of 37% of fish phenotypes wrongly classified as the opposite sex across 100 RF simulations. While the RF analysis of error rate indicated that the 2 Mb tested on Omy1 were really useful to discriminate the dams with high and low progeny sex-reversal ratios, the median error rate derived from the RF analysis on Omy12 (152 haplotypes spanning 8.5 to 9.5 Mb) was 42% (ranging from 29% to 51% across the 100 RF runs), indicating that the best haplotypes and genes identified on this chromosome were not so strongly associated with sex-reversal. In terms of error rate, the RF analysis on the 707 haplotypes covering the QTL regions on Omy20 (247 haplotypes from 27 to 29 Mb, 288 haplotypes from 33.5 to 35.5 Mb, and 172 haplotypes from 36 to 37 Mb) gave intermediate results in-between the two previous analyses, misclassifying one third of the dams on average with a minimum error rate of 29% and a maximum rate of 38% across 100 RF simulations.

These results are consistent with the higher proportions of genetic variance explained by QTL on Omy1 than on Omy12 or Omy20 [30]. In addition, the 1,105 haplotypes covering the QTL regions along the three chromosomes were considered all together in a subsequent full RF analysis. The empirical distribution of error rates for the full RF analysis gave median, minimum and maximum values similar to the ones of the RF on Omy20 (33%, 29%, and 38%, respectively) while we were expecting that these values will be at least as low as for the RF on Omy1. Due to the very small size of our dataset (n = 23), it is likely that having over 1,100 haplotypes to consider in the full RF analysis created some "noise" in the classification process.

The top ten RF ranked haplotypes in the 2 Mb evaluated on Omy1 (see Table E in S1 Tables) fell within the nine following genes (Fig.2): *sdhaf4*, *tlx1*, *syndig1* and *vsx1* genes located in QTL Omy1_a; *cep68*, *fbxw4*, *hells*, LOC110527930, and *gbf1* in QTL Omy1_b. Six of those genes have been identified with significant effects on sex-reversal in several validation populations (Table 2) or within some significant SNPs in the discovery population (Table 4). The first top-ranked haplotype included two genes from QTL Omy1_b (*sdhaf4* and *tlx1*), while the second and third top-ranked haplotypes were in the sequence of the *syndig1* gene on Omy1_a.

**Table 3.** Redefined boundaries of sex-reversal QTL regions in the current study (location on Arlee reference genome) compared to QTL intervals reported by Fraslin et al. [30].

| Chromosome | Region name | QTL name | Start of QTL (bp) | End of QTL (bp) | Old boundaries on Swanson reference* |
|---|---|---|---|---|---|
| Omy1 | Z1 | Omy1_a | 67 000 061 | 67 960 401 | 63 229 533–63 560 537 bp |
| Omy1 | Z1 | Omy1_b | 67 960 402 | 68 999 919 | 64 354 655–64 707 100 bp |
| Omy12 | Z2 | Omy12 | 8 500 067 | 9 499 932 | 5 751 451 –6 965 249 bp |
| Omy20 | Z3 | Omy20_a | 27 000 149 | 28 998 983 | Around |
| Omy20 | Z4 | Omy20_b | 33 508 998 | 35 499 146 | 31 352 390 bp |
| Omy20 | Z5 | Omy20_c | 36 000 252 | 36 999 928 | |

*Swanson reference assembly genome: Omyk_1.0 (GenBank assembly accession GCA_002163495.1)

**Table 4.** Most significant SNPs by Fisher's exact test (p_value ≤ 0.002) considering the 23 dams with extreme progeny sex-ratios in the QTL discovery population.

| Omy | Position | REF | ALT | p-value | NCBI gene annotation (Genecards symbol) |
|---|---|---|---|---|---|
| 1 | 67931300 | A | C | 0.0017 | opsin 4a (ops4a) |
| 1 | 68330279 | C | T | 0.0017 | 8.758 kb upstream of F-box and WD repeat domain containing 4 (fbxw4) |
| 1 | 68832468 | G | A | 0.0013 | Golgi-specific brefeldin A-resistance guanine nucleotide exchange factor 1 (gbf1) |
| 20 | 27841889 | A | G | 0.0020 | ARFGEF family member 3 (arfgef3) |
| 20 | 27841897 | T | A | 0.0020 | ARFGEF family member 3 (arfgef3) |
| 20 | 28044820 | G | C | 0.0017 | v-akt murine thymoma viral oncogene homolog 3a (akt3a) |
| 20 | 28206089 | A | G | 0.0017 | SHH signaling and ciliogenesis regulator sdccag8 (sdccag8) |
| 20 | 28224422 | C | *,T | 0.0020 | SHH signaling and ciliogenesis regulator sdccag8 (sdccag8) |
| 20 | 28270362 | A | G | 0.0017 | centrosomal protein 170aa (cep170aa) |
| 20 | 28333748 | G | T | 0.0017 | 0.117 kb upstream leucine-rich pentatricopeptide repeat containing (lrpprc) |
| 20 | 28695801 | A | G | 0.0006 | calmodulin-lysine N-methyltransferase (camkmt) |
| 20 | 28775897 | A | T | 0.0013 | 30.316 kb downstream calmodulin-lysine N-methyltransferase (camkmt) |
| 20 | 28803467 | A | C,T | 0.0005 | 45.584 kb upstream SIX homeobox 3a (six3a) |
| 20 | 33670075 | G | A,T | 0.0017 | KH domain-containing, RNA-binding, signal transduction-associated protein 2 (khdrbs2) |
| 20 | 34018207 | C | T | 0.0006 | Dystonin (dst) |
| 20 | 35051101 | G | T | 0.0017 | CUB and sushi domain-containing protein 1 (csmd1) |
| 20 | 36030963 | G | T | 0.0013 | 0.468 kb upstream uncharacterized ncRNA (LOC110499423) |
| 20 | 37557112 | C | G | 0.0006 | ENSOMYG00000068776* |
| 20 | 37995993 | T | G,* | 0.0017 | 79.980 kb upstream charged multivesicular body protein 6 (chmp6) |

* ENSOMYG00000068776 on Omy20 (37,056,423... 37,561,145) is a paralog of rbfox3 located on Omy23

These results are not fully consistent with our first GWAS [30] using progeny imputed genotypes and Bayesian mixed model methodologies. In this previous analysis, the *syndig1* gene was not included in the credibility interval for Omy1_a. Although there was no SNP within this gene among the most significant SNPs (p_value ≤ 0.002) in the discovery population, two SNPs had, however, a p_value < 0.005 (see Table D in S1 Tables). In addition, *syndig1* exhibited highly significant SNPs in four validation populations, including population A. These results make *syndig1* a very plausible candidate gene for QTL Omy1_a. As regards to the initial QTL Omy1_b, there is now an extended credibility interval, including genes before *slc2a15a* such as *tlx1* and *cep68*, and genes after the uncharactezized LOC110527930 such as *gbf1*. While LOC110527930 initially was the best candidate gene for Omy1_b, the top-ranked haplotypes now point towards genes such as *tlx1*, *cep68*, *fbxw4* and *hells* that are now prefered within Omy1_b region (Fig. 2).

On the 1 Mb tested on Omy12 (see Table F in S1 Tables), the first three haplotypes most likely explain sex-reversal were within the same two *hcn1* and *ccbe1* genes that were also identified in three populations unrelated to the discovery population (see Table C in S1 Tables).

On Omy20 (Fig.3) the best ten ranked haplotypes in a 5Mb window investigated in the RF analysis (see Table G in S1 Tables) were overlapping with the following eight genes, presented in decreasing order of importance: *lrrc59*, *csmd1*, *khdrbs2*, *abcg5*, *akt3*, *rgs17*, *caskin2* and *dst*. No variants from this region could be tested in the validation populations, however, some highly significant SNPs are detected in the discovery population for four of these genes (*akt3*, *khdrbs2*, *dst* and *csmd1*, see Table 4).

When all the haplotypes were analyzed together across all the QTL regions spanning the 3 chromosomes (see Table H in S1 Tables) and ranked through RF, the first one was positionned within *lrrc59* gene on Omy20_c, the second and

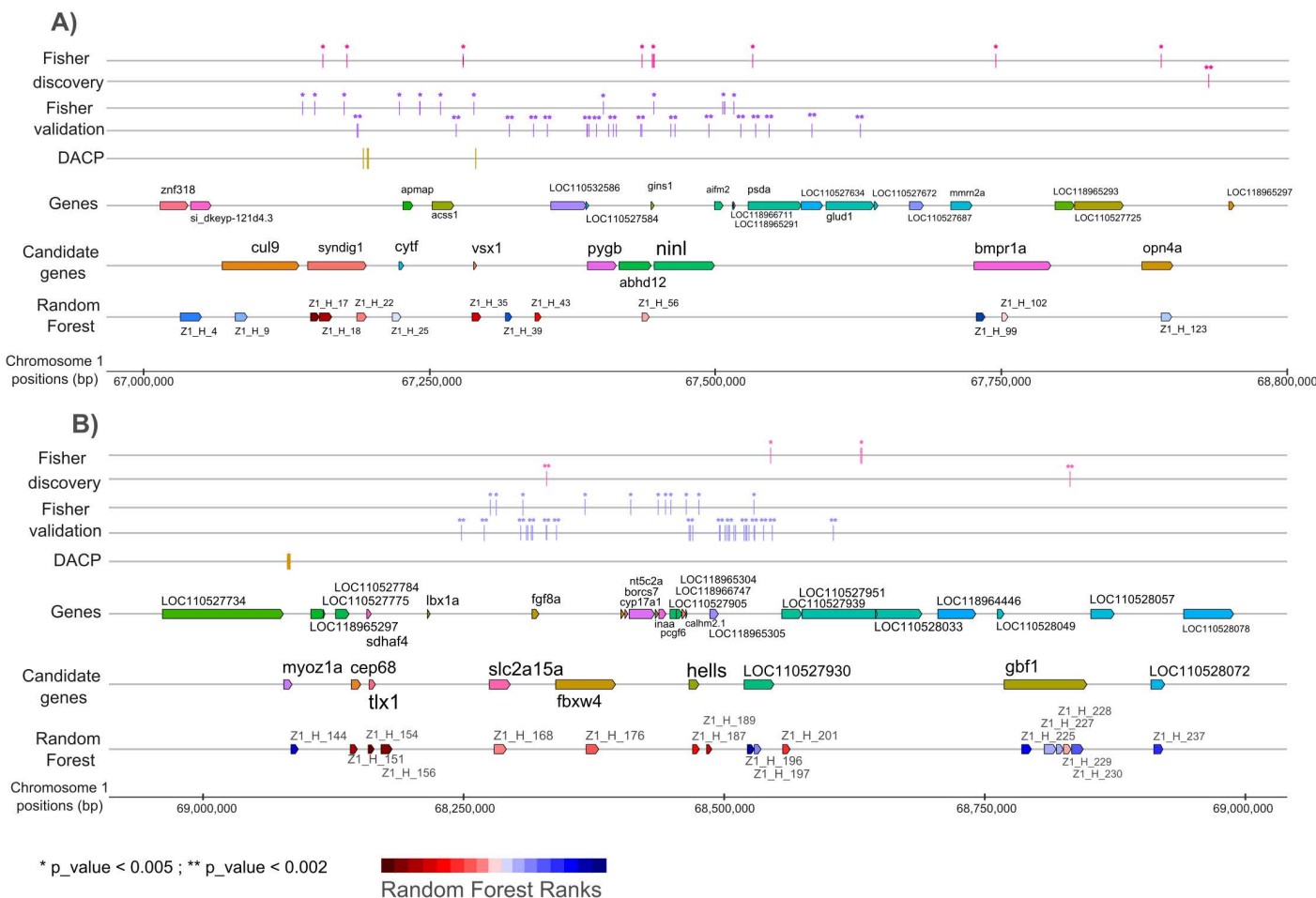

* p_value < 0.005 ; ** p_value < 0.002
Random Forest Ranks

**Fig. 2. Positions of genes within QTL regions a and b on Omy1 with significant SNPs positions in discovery or validation population as well as best ranked haplotypes.** A) QTL region a on chromosome Omy1. B) QTL region b on chromosome Omy1. Results of the Fisher exact test in the Discovery population (Fisher Discovery, in pink) and the validation populations (Fisher Validation, in purple) and of the Discriminant Analysis of Principal Components (DAPC) in the discovery population (in yellow). Each bar represents a SNP that was tested for association (* p_value < 0.005; ** p_value < 0.002). All genes and candidate genes (based on the combination of test performed in this manuscript) are presented along the QTL. Haplotypes are ranked using the Mean Gini Index value over 100 runs of Random Forrest analysis performed within chromosome Omy1 (dark red: highest Gini index to dark blue: lowest Gini index, values available in Table E in S1 Tables).

third ones included *sdhaf4* and *tlx1* on Omy1_b, and the fourth one was located within the *csmd1* gene on Omy20_b. The next four top-ranked haplotypes were all on Omy20, located within *abcg5*, *caskin2*, *khdrbs2*, and *dst* genes. Some genes identified on Omy1 in validation populations (*syndig1*, *hells*, *fbxw4* and LOC110527930) appeared inn the RF haplotype analyses ranked with very similar Gini indices (Fig 2). Noticeably, most of the top-ranked haplotypes invovled sequences located within genes, and the ranking was consistent between the full RF analysis and the RF limited to their chromosomic location.

Based on the results of the 60 top-ranked haplotypes on the Gini index of this full RF analysis, as well as the additional results from Fisher's exact tests (Tables 2 and 4), we identified a list of 45 genes (see Table I in S1 Tables) whose variants in their sequence or in their close vicinity (2kb) were the most likely to explain a significant part of sex-reversal phenotypes.

### Identification of the potential causative variants combining DAPC and SNP annotations

In order to find the best 100 SNPs to discriminate among the 23 dams the ones with high progeny sex-reversal ratios from those with low ratios, the sequence variants of those 45 genes (±2 kb) were considered in a DAPC analysis.. Running a simple PCA on all those 27,828 variants (without missing genotypes out of 33,731 SNPs) did not allow to correctly discriminate the two groups of dams (Fig 4A) whereas the DAPC perfectly discriminated the two groups (Fig 4B).

The genotypes for the 100 top SNPs (from 27,828, SNPs) that were genotyped for the 23 dams, as well as for five additional dams, are given in Table J in S1 Tables. These five dams, with at least six offspring with a record for phenotypic sex the discovery population (see Table A in S1 Tables), were used as an internal validation: three dams (noted AS, AT and AU) with high progeny sex-reversal ratio and two dams (noted BS and BT) with no sex-reversed offspring. The top-ranked 100 SNPs were located within or nearby the downstream or upstream regions of 16 genes belonging to the list of the 45 genes previously identified by RF and Fisher's exact test: on Omy1, *syndig1*, *vsx1*, and *myoz1*; on Omy12, *hcn1* and *ccbe1*; on Omy20, *arfgef3*, *cep170aa*, *abcg5*, *khdrbs2*, *dst*, *hmgcll1*, *cilk1*, *rps6ka2*, LOC110499423, *lrrc59* and *caskin2*. Most of these 100 SNPs were located in intronic regions of the genes, two were synonymous mutations and none were splice donor variants in introns or missense variants in exons. When focusing on the 50 best SNPs to discriminate between the two groups of dams, only *hcn1* remained on Omy12; *syndig1* and *myoz1* were kept for Omy1_a and Omy1_b, respectively; and, all genes, except LOC110499423, remained for QTL regions on Omy20.

## Discussion

### Significance of the different QTL regions across French rainbow trout populations

All the QTL regions previously identified [30] were confirmed as associated with female to male sex-reversal in two or more populations different from the discovery one. Using the list of the 100 top-ranked SNPs under DAPC in the discovery population we confirmed the hypothesis of two QTL on Omy1, one QTL on Omy12, and we now suggest the existence of three distinct QTL on Omy20.

Regarding QTL Omy1_a, the most relevant candidate genes to explain sex-reversal across populations were *syndig1*, *acss1*, *vsx1*, *entpd6*, *pygb* and *ninl*. In the initial study [30], the peak SNPs for this QTL were located either within the *pygb* or *ninl* gene, depending on the statistical approach. In particular, Fraslin et al. [30] mentioned a SNP located at 63,543,061 bp (on Swanson reference genome) within the *ninl* gene and annotated has a missense variant. This SNP was not genotyped in the validation population, but in this study, a very close SNP located at 63,530,416 bp on Swanson reference genome (corresponding to 67,434,621 bp on Arlee reference genome) has a putative effect in population A and a significant effect at the genome level in populations B and C (see Table C in S1 Tables).

In our previous study [30], all peak SNPs from the main QTL Omy1_b were located in the intergenic region between the *hells* gene and the uncharacterized protein LOC110527930, the latter being proposed as the most relevant positional candidate for explaining 14% of the genetic variance of sex-reversal in the discovery population. However, in the present

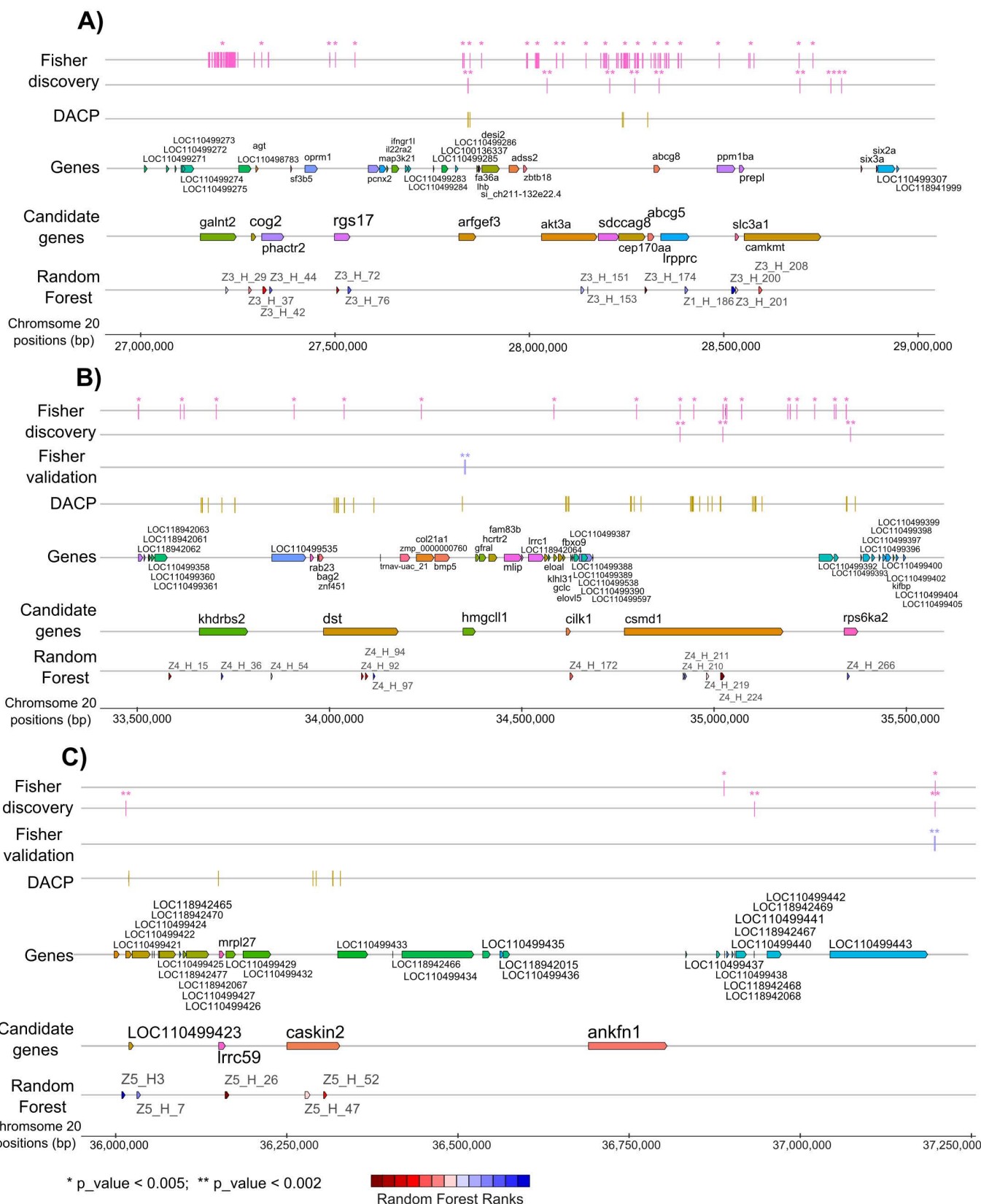

**Fig. 3. Positions of genes within QTL regions a, b and c on Omy20 with significant SNPs positions in discovery or validation population as well as best ranked haplotypes.** A) QTL region a on chromosome Omy20. B) QTL region b on chromosome Omy20. C) QTL region c on chromosome

Omy20. Results of the Fisher exact test in the Discovery population (Fisher Discovery, in pink) and the validation populations (Fisher Validation, in purple) and of the Discriminant Analysis of Principal Components (DAPC) in the discovery population (in yellow). Each bar represents a SNP that was tested for association (* p_value < 0.005; ** p_value < 0.002). All genes and candidate genes (based on the combination of test performed in this manuscript) are presented along the QTL. Haplotypes are ranked using the Mean Gini Index value over 100 runs of Random Forrest analysis performed within chromosome Omy20 (dark red: highest Gini index to dark blue: lowest Gini index, values available in Table G in S1 Tables).

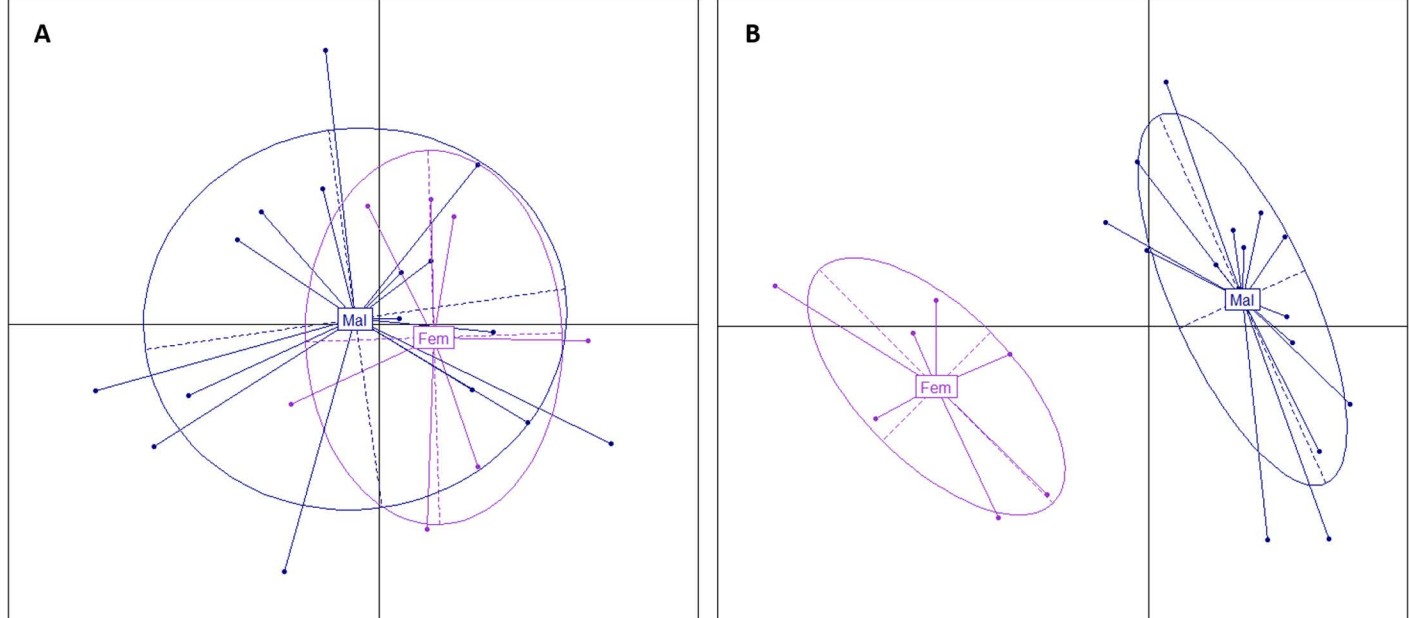

**Fig 4. PCA of variants between "Fem" and "Mal" dam groups on 45 candidate genes.** A. PCA of the 27,828 variants in the 45 candidate genes (±2 kb) and B. PCA of the 100 top variants selected by DAPC to discriminate the "Fem" group of the seven dams with low progeny sex-reversal ratios and the "Mal" group of the 16 dams with high progeny sex-reversal ratios.

study, other genes should also be considered as candidate genes in several validation populations. Some were located before the *hells* gene (*slc2a15a*, *fgf8*, and *fbxw4*), while the most distal one, *col13a1*, was located after LOC110527930. This observation, as well as the identification of *syndig1* in QTL Omy1_a that was positioned before the first QTL interval defined in Fraslin et al. [30] were the motivations to enlarge the genomic region investigated in the current study.

In the previous study [30], the QTL on Omy12 was detected using the 57K chip, but not when using imputed sequence variants in the discovery population. In the present study, this QTL is validated in population D and F with a significant SNP identified within *hcn1* gene but is not validated in population A although it was closely related to the discovery population. We can hypothezise that the QTL is segregating in population A, but either at a too low frequency or too small effect to be statistically detected.

The QTL region on Omy20 was not validated in population A, but two QTL are segregated on Omy20 in populations C, D and F, with significant SNPs located in two distant genes: *hmgcll1* located around 34.4 Mb, and ENSOMYG00000068776 gene located around 37.5 Mb. The same hypothesis as for the QTL on Omy12 can thus be proposed for the absence of statistical validation in population A. In addition, in an INRAE experimental rainbow trout line, a QTL for spontaneous sex-reversal was also identified on Omy20 (previously named RT17) but with a very imprecise position [31].

In the next paragraphs, we describe the putative functional roles of our top positional candidate genes for the two QTL on Omy1 (Fig 2) as well as for the most promising genes for the three different QTL on Omy20 (Fig 3). Other positional candidate genes on Omy12, Omy20_a and Omy20_b with less functional evidence are presented in an additional section [see S1 Appendix].

### *syndig1* as the best positional and functional candidate gene for QTL Omy1_a

*syndig1* encodes a protein that belongs to the interferon-induced transmembrane family of proteins and plays a critical role during synapse development as well as in postsynaptic development and maturation [51,52]. Although *syndig1* expression is mainly restricted in the embryonic retina in mice [53], its expression has been detected also in granulosa cells at the small antral stage in sheep [54] suggesting a potential role in follicular development in vertebrates. It is most likely that one or a few variants in the *syndig1* (synapse differentiation-inducing gene protein 1) sequence or in its close downstream region (< 1kb) are the causative variants explaining part of the sex-reversal phenotype. Indeed, four SNPs were significant in the Fisher's exact test in two to four validation populations. In the discovery population, the RF analysis highlighted three haplotypes in this gene among which three missense variants were annotated, and, finally, the DAPC ranked three SNPs within or downstream of *syndig1* among the 100 best SNPs to discriminate the low and high sex-reversed progeny dams (see Tables C, E, and J in S1 Tables). However, among the 50 best SNPs, the only two retained were located in downstream intergenic positions. We speculate that these latter positions were kept to the detriment of SNPs within *syndig1* due to linkage disequilibrium between SNPs of the two close genes *syndig1* and *vsx1* (visual system homeobox 1).

A unique SNP was retained from *vsx1* in the list of the 100 best SNPs and was filtered out in the list of the 50 best ones. We may speculate that *vsx1* may play a minor additional role to explain sex-reversal as significant SNPs (including five missense variants) were detected in the discovery population and in four validation populations. However, as far as we know, no direct link has been suggested between *vsx1* and sex differentitation in any species.

### Multiple candidate genes for QTL Omy1_b, with *tlx1* and *hells* as best functional candidates

While this QTL was identified as the one explaining the largest proportion of genetic variance of spontaneous sex-reversal [30], our previous study pointed on different candidate genes than in the current one. In our first study, the three putative functional candidate genes were *fgf8a*, *cyp17a1* and LOC110527930 based on progeny imputed genotypes, while the current analysis of dams' sequences using an updated genome assembly pointed to genes located upstream, prioritizing *tlx1* (T cell leukemia homeobox 1) for RF analysis and *myoz1* (myozenin-1) for DAPC. It seems, in fact, that two different QTL should be distinguished within Omy1_b, Omy1_b1 and Omy1_b2, on each side of *cyp17a1*. The new analyses highlighted *myoz1* and *tlx1* that are located in Omy1_b1 before the start of the QTL region previously defined [30]. Although *cyp17a1* is known to be involved in gonad masculinization in the common carp and zebrafish [55,56], no variant in this gene was significantly associated with sex-reversal in rainbow trout, neither in the discovery nor in the validation populations. However, it is possible that a cis-regulatory element within the QTL region could control, at long-distance, the expression of *cyp17a1*, influencing sex differentiation in rainbow trout. Such long-distance cis-regulatory elements have been found to control sex like in the Sry-deficient Amami spiny rat where the male-specific upregulation of *sox9* is controlled by an enhancer region 0.5Mb upstream of *sox9* [57].

The initial Omy1_b from [30] is now defined as a second genomic region Omy1_b2, spanning from *fbxw4* to *gbf1* (including *hells* and LOC110527930). The RF analysis identified some haplotypes among the 10 best ones to explain sex-reversal on Omy1, putting similar emphasis to four genes: *fbxw4*, *hells*, LOC110527930 and *gbf1*. In addition, Fisher's exact test in the discovery population identified a significant SNP within *gbf1* (Golgi-specific brefeldin A-resistance guanine nucleotide exchange factor 1). SNPs in this gene were not tested in the validation populations as outside the limits of the initial boundaries for the QTL [30]. *gbf1*, is one of the guanine-nucleotide exchange factors (GEF) for members of the Arf

family of small GTPases involved in vesicular trafficking in the early secretory pathway (see Table I in S1 Tables). A *gbf1* expression was shown throughout spermatocyte development in the Golgi apparatus with expression extending into early spermatids where it localizes also in the acrosome [58]. No direct link with gonad development, oogonia or spermatogognia could be established for *fbxw4* (F-box and WD repeat domain containing 4). This gene is a member of the F-box/WD-40 gene family that may participate in Wnt signaling which is a pathway involved in gonadal differentiation (see Table I in S1 Tables). The *hells* (helicase lymphoid specific) gene (also known as *lsh*) is a very convincing functional candidate gene for sex-reversal. It is a member of the *snf2* helicase family that is required for normal development and survival in the mouse [59], and is implicated in the control of genome-wide DNA methylation [60]. Analysis of ovarian explants obtained from *hells* mutant females demonstrates that lack of *hells* function is associated with severe oocyte loss and lack of ovarian follicle formation [61]. Although *hells* is expressed in undifferentiated embryonic stem cells, it is significantly downregulated upon differentiation [62]. Using allografting of testis tissue from *hells* -/- mice to study postnatal male germ cell differentiation, Zeng et al. [63] showed that proliferation of spermatogonia was reduced in the absence of *hells*, and germ cell differentiation arrested at the midpachytene stage, implicating an essential role for *hells* during male meiosis.

Although Omy1_b1 was a strong candidate identified in both RF and DAPC analyses, there is no direct evidence for a potential role of *myoz1* gene in sex-reversal (see Table I in S1 Tables). The second-best candidate gene *tlx1* (also known as *hox11*), encodes a nuclear transcription factor that belongs to the NK-linked or NK-like (*nkl*) subfamily of homeobox genes, essential for cell survival during spleen development in fish [64] and mammals [65,66]. It is also involved in specification of neuronal cell fates during embryogenesis (see Table I in S1 Tables), and *tlx1* can either activate or repress gene transcription [67]. *tlx1*-dependent regulation of retinoic acid (RA) metabolism is critical for spleen organogenesis [68]. RA is the active metabolite of vitamin A that is required for vertebrate embryogenesis [69,70]. In mice, the transcription factor steroidogenic factor 1 (*nr5a1* also known as SF1) regulates RA metabolism during germ cell development [71] and is essential for sexual differentiation and formation of the primary steroidogenic tissues. Indeed, mutations within *nr5a1* underlie different disorders of sexual development, including sex reversal. In patients with disorders of sexual development, a mutant form of *nr5a1* was shown to be defective in activating *tlx1* transcription [72]. Thus, it can be hypothetized that genetic variant in *tlx1* may also directly deregulate RA metabolism, inducing sex-reversal.

Gathering all the information presented here for the QTL on Omy1, we concluded that there is a strong statistical and biological evidence of significant involvment in the sex determination and differentiation cascade of *syndig1*, *tlx1*, *hells* and *gbf1*, and, putatively also of *vsx1*, and *myoz1*. Notably, at the only exception of *myoz1*, all these proteins are relatively well conserved in terms of protein identity across various fishes and mammals (See Table K in S1 Tables).

### *arfgef3* as the best functionnal candidate for Omy20_a due to its activation of the estrogen/ERα signaling pathway

*arfgef3* (ARFGEF family member 3) gene (named also *big3*) is a member of the *big1*/*sec7p* subfamily of ADP ribosylation factor-GTP exchange factors (ARF-GEFs). The protein is very well conserved across vertebrates (see Table K in S1 Tables). *arfgef3* overexpression has been identified as one of the important mechanisms causing the activation of the estrogen/ERα signaling pathway in the hormone-related growth of breast cancer cells [73]. Control of estradiol synthesis could play a key role not only for ovarian, but also for testicular differentiation and sex change in fish [12].

### *khdrbs2*, *dst*, *hmgcll1* and *csmd1* as functional candidates for QTL Omy20_b

*Implication of khdrbs genes in sex determination and regulation of alternative splicing.* In nematodes, *gld*-1, the ortholog of *khdrbs* (KH domain-containing, RNA-binding, signal transduction-associated protein 2), is indispensable for oogenesis and meiotic prophase progression and it can stimulate sex determination of males in the hermaphrodite germ line [74, 75, 76]. In fruit fly, *nsr*, the ortholog of *khdrbs* regulates some male fertility genes in the primary [77]. In vertebrates, *khdrbs* (*khdrbs1*, *khdrbs2* and *khdrbs3*) genes encode respectively for the SAM68, SLM1 and SLM2 proteins

that have RNA binding and signal transduction activities [78,79]. In fish, the *khdrbs1* was further duplicated through the teleost-specific whole genome duplication, forming *khdrbs1a* and *khdrbs1b* [80]. In zebrafish, all the *khdrbs* genes were found to be primarily expressed in the brain during early development and at the adult stage. *khdrbs1a* was also found to be expressed in the gonad primordium as well as in the gonads of adult fish along with *khdrbs1b*, and *khdrbs3* [80]. In our study, eight intronic *khdrbs2* variants were ranked among the 100 top SNPs in DAPC (see Table J in S1 Tables), and an indel/SNP variant was identified with a p-value < 0.002 in the discovery population (Table 4). In mouse, *khdrbs1* and *khdrbs3* are predominantly expressed in the brain and testis, while *khdrbs2* is exclusively expressed in the brain [81]. *khdrbs2* knockout mice are fertile whereas *khdrbs1* knockout mice show impaired fertility in males [81]. *khdrbs3* is involved in spermatogenesis via directly binding to RNA [81]. In addition, *khdrbs1* knockout females displayed a reduction in the number of developing ovarian follicles, alteration of estrous cycles, and impaired fertility (e.g., [82]). It seems that the invertebrate genes *gld-1* and *nsr* that are orthologs of *khdrbs1* and *khdrbs3*, respectively, have similar roles in the regulation of gametogenesis [80].

### dst and hmgcll1 as novel players in sex determination and differentiation in fish?

*dst* (dystonin) encodes a member of the plakin protein family of adhesion junction plaque proteins. Low temperatures significantly impact *dst* splicing in killifish, stickleback and zebrafish [83]. As far as we know, *dst* has not yet been demonstrated as being involved in sex determination or differentiation. Nevertheless, there is no statistical doubt that this gene is invoved in sex-reversal in rainbow trout. A SNP was highly significant (p-value = 0.0006) in the discovery population (Table 4) with homozygous TT dams exhibiting high ratios of progeny sex-reversal (see Table J in S1 Tables). In addition, four haplotypes within the gene ranked among the 60 top in the full RF analysis (see Table H in S1 Tables), and 15 intronic SNPs were ranked among the 100 top in DAPC, 10 remaining in the 50 top SNPs (see Table J in S1 Tables).

*hmgcll1* (3-hydroxymethyl-3-methylglutaryl-CoA lyase-like 1) was first characterized as a lyase activity enzyme localized in extra-mitochondrial region involved in ketogenesis for energy production in nonhepatic animal tissues (see Table I in S1 Tables). A novel role of *hmgcll1* in cell cycle regulation was recently suggested [84] that could be linked with a putative functional role on sex-reversal. Statistically, there is no doubt about the effect of this gene on sex-reversal in rainbow trout. Two SNPs within the gene were significant in two validation population (Table 3). In the discovery population, an upstream intergenic variant at position 34,345,558 bp was among the 50 top-ranked in DAPC, with homozygous TT dams exhibited high ratios of progeny sex-reversal (see Table J in S1 Tables).

### csmd1 is associated to gonadal failure in mammals

The *csmd1* (CUB and sushi multiple domains protein 1) gene was one of the best positional candidates in our RF and DAPC analyses with 8 SNPs being among the 50 top ones to discriminate the two groups of extreme phenotypes. The family of proteins *csmd* is one of the three types of the complement system-related proteins involved in the recognition of molecules in innate immune-system and in the central nervous system and *csmd1* is predicted to act on several processes, including reproductive structure development (reviewed in [85]). The large transmembrane protein encoded by *csmd1* is highly conserved between vertebrates with 89% amino acid sequence identity between trout and medaka and still 77% between trout and human or mouse (see Table K in S1 Tables). In humans and mice, *csmd1* was associated with gonadal failure in both sexes [86]. Significant association between early menopause and rare 5′-deletions of *csmd1* was detected in humans. In mice, knockout females show significant reduction in ovarian quality and breeding success, and *csmd1*-knockout males show increased rates of infertility and severe histological degeneration of the testes. No variants predicted with moderate or high impacts on protein expression in this gene were observed. But, as indicated by Lee at al. [86], intronic variants mainly located in introns 1 and 2 likely harbored functional elements that can influence *csmd1* expression in testis.

### *lrrc59* and *caskin2* as the best positional candidates for QTL Omy20_c

As far as we know, no direct effect of *lrrc59* (leucine-rich repeat-containing protein 59) on sex determination is known. This gene enables RNA binding activity and cadherin binding activity and plays a key role in the regulation of local translation in the endoplasmic reticulum [87] (see Table I in S1 Tables). Statistically, it was the top-ranked haplotype on the full RF analysis (see Table H in S1 Tables), indicating a strong importance of *lrrc59* to predict sex-reversal.

   *caskin2* (CASK Interacting Protein 2) downregulates genes associated with endothelial cell activation and upregulates genes associated with endothelial cell quiescence [88]. Morpholino knockdown of *caskin2* in zebrafish results in abnormal vascular development, while *caskin2* knockout mice are viable and fertile [88]. *caskin2* is not well conserved across vertebrates, including only 54% AA identity between rainbow trout and either Medaka or Zebrafish (see Table K in S1 Tables). Statistically, two haplotypes including parts of *caskin2* were among the top-ranked haplotypes in the full RF analysis (see Table H in S1 Tables). In addition, three SNPs were among the 50 top- ranked in DAPC (see Table J in S1 Tables). The expression of *caskin2* was down-regulated in gonads of *Gobiocypris rarus* of both sexes, when exposed to a 7-day exposure to 17α-methyltestosterone which had a sex-reversal effect [89]. In Coho salmon, androgens play a major role in stimulating primary ovarian follicle development and in the transition into secondary follicles, in particular inducing a differential expression of *caskin2* after 3-day treatment with non-aromatizable androgen 11-ketotestosterone [90].

### Putative role of *foxl3* in germ cell fate decision

Intriguingly, *foxl3* (Forkhead Box L3), a co-paralog gene of *foxl2*, is located on Omy20 spanning from 22,942,515–22,948,650 bp, upstream the QTL regions we studied on the same chromosome. Although *foxl3* is expressed in the gonads of teleosts, no variant within *foxl3* were associated to sex-reversal in our study. However, we can speculate that *foxl3* is involved in germ cell fate decision in rainbow trout. Indeed, *foxl3* has been suggested to be involved in the onset of oocyte meiosis, and/or in the regulation of male specific genes during late testis development and testis maturation in salmonids [91, 92, 93]. It was also demonstrated in medaka that *foxl3* expression is only maintained in female germ cells, and that functional sperm is produced in the ovary of a *foxl3* loss-of-function mutant [94]. In addition, *foxl3* was shown to initiate oogenesis in medaka through two genetically independent pathways, meiosis and follicular development, and that a third pathway indepentdent of *foxl3* may also be hypothetized [95]. Last, *foxl3* expression levels were strongly increased in gonads during the natural female-to-male sex reversal in the rice field eel (*Monopterus albus*). *foxl3* seemed to be involved in physiological processes that promote oocyte degeneration in the ovotestis and stimulating spermatogenesis in spermatogonia [96]. Therefore, we can hypothetize that expression of *foxl3* may be regulated by some other genes, presumably located in the QTL regions identified on Omy20 in our study, similarly to the long-distance cis-regulation observed for *sox9* expression [57]. Further investigation should aim at validating such speculations, as well as the causative variants and precise functional roles of the identified genes in our QTL regions.

### Interaction effects between spontaneous masculinization and rearing temperature in farming and ecological contexts

In addition to the genetic influence, it is well known that water temperature during development can modulate sex differentiation in many fish species and, in particular, high temperature is associated with masculinization [97]. In rainbow trout, high rearing temperature during the early phase of fry development after hatching was shown to increase the frequency of sex reversal in XX females to a greater or lesser extent depending on the genetic background of the trout lines [98,99]. From a fish farmer perspective, further investigation is still needed before any operational use of the genotypes at minor sex-modifier genes either to eradicate spontaneous sex-reversal in all-female production stocks or to produce progeny from free-hormone neomales in rainbow trout. Indeed, the high heritability of spontaneous maleness in XX individuals [30] suggests that the use of spontaneously masculinised individuals as progenitors of all-female stocks would increase the frequency of undesirable masculinised progeny. Improved knowledge about the joint effects of genetic basis and

environmental factors determining spontaneous maleness in all-female stocks is needed to search for a trade-off combining together genetic and environmental control of gonad masculinization according to the destination of the fish (broodstock vs all-female production stock).

From an ecological point of view, the question of the extent of spontaneous sex reversal arises as a key one for conservation biology and wild fish population sex ratios in a global warming context [100]. A study on zebrafish [100] compared the impact of elevated-temperature on sex reversal in different strains with chromosomal or polygenic GSD. They showed that strains with a major sex-determining locus may not be immune to temperature driven sex reversal. Spontaneous sex reversed females in chinook salmon (*O. tshawytscha*), Nile tilapia (*Oreochromis niloticus*), zebrafish (*Danio rerio*) and lake trout (*Salvelinus namaycush*), have been observed at low frequency in the wild [1,25,100–103]. The impact of spontaneous masculinization in wild rainbow trout on population dynamics and fitness remains unknown and deserves more attention as this highly heritable phenomenon, driven by environmental factors, may become more frequent with warmer water temperature.

## Conclusions

In the context of global warming, it is essential to better understand the genetic by environment interactions controlling spontaneous masculinization of rainbow trout, both in farmed environment but also in wild populations. While considerable scientific effort has been made over the last fifty years to unravel mechanisms for sex-determination and differentiation, the nature of genetic variations that regulate these mechanisms in fish is still not well-known. We confirmed on several French commercial rainbow trout populations the importance of some genomic regions on chromosomes Omy1, Omy12 and Omy20 for spontaneous masculinization of XX-individuals. Those regions can be considered as minor sex-determining regions in rainbow trout. Together, our results are consistent with a model in which spontaneous female-to-male sex-reversal in rainbow trout is associated with genetic factors able to reduce germ cell proliferation and arrest oogenesis. The main candidate genes that we suggested based on positional and functional information are *syndig1*, *tlx1*, *hells* and *gbf1* on Omy1, and *arfgef3*, *khdrbs2*, *dst*, *hmgcll1*, *csmd1*, *lrrc59* and *caskin2* on Omy20. Most of these genes were previously indicated as playing roles in developmental pathways in vertebrates.

## Supporting information

**S1 Appendix.**   Positional candidate genes on Omy12 and Omy20 without known evidence of a functional role linked to sex-reversal
(DOCX)

**S1 Tables.**   This excel file includes 11 supplementary tables whose titles are listed below. Table A. Number of progeny (female, intersex and male) and proportion of neomales per sequenced dam. Table B. Positions on the Swanson reference genome and flanking sequences of the 192 SNPs used in validation. Table C. List of the 192 SNPs detected in 6 validation populations (named A to F) with their positions on Swanson (SW) and Arlee (AR) reference genomes. Table D. List of most significant SNPs found in Omy1, Omy12 and Omy20 QTL. Table E. Mean, Minimum and Maximum values of the 30 haplotypes with the highest Gini index over 100 runs of Random Forests for Omy1 QTL. Table F. Mean, Minimum and Maximum values of the 10 haplotypes with the highest Gini index over 100 runs of Random Forests for Omy12 QTL. Table G. Mean, Minimum and Maximum values of the 30 haplotypes with the highest Gini index over 100 runs of Random Forests for the 3 Omy 20 QTL (Z3, Z4, Z5). Table H. Mean values of the 60 haplotypes with the highest Gini index over 100 runs of Random Forests for the QTL regions of Omy1 (Z1), Omy12 (Z2) and Omy20 (Z3 to Z5). Table I. List of potential candidate genes for QTL on Omy1, Omy12 and Omy20 with their descriptions on GeneCards, UniProtKB/Swiss-Prot and NCBI databases. Table J. Location, annotation, alleles and genotypes for sequenced dams of the 100 most important SNPs in DAPC analysis. Genotypes in blue are considered as male genotypes, genotypes in orange as female genotypes

and genotypes in light orange are heterozygous. Table K. Protein sequence identities of genes associated to sex-reversal in rainbow trout across six species in Vertebrates.
(XLS)

## Acknowledgements

The authors are grateful to the companies "Les Fils de Charles Murgat", "Aqualande", "Bretagne Truite", "Font Rome" and "Viviers de Sarrance" for fish samples and data collection.

## Author contributions

**Conceptualization:** Edwige Quillet, Florence Phocas.

**Data curation:** Audrey Dehaullon, Clémence Fraslin.

**Formal analysis:** Audrey Dehaullon, Clémence Fraslin.

**Funding acquisition:** Edwige Quillet.

**Investigation:** Florence Phocas.

**Methodology:** Florence Phocas.

**Project administration:** Edwige Quillet, Florence Phocas.

**Resources:** Anastasia Bestin, Charles Poncet, Yann Guiguen, Edwige Quillet.

**Supervision:** Florence Phocas.

**Validation:** Yann Guiguen, Florence Phocas.

**Visualization:** Audrey Dehaullon, Clémence Fraslin.

**Writing – original draft:** Florence Phocas.

**Writing – review & editing:** Audrey Dehaullon, Clémence Fraslin, Anastasia Bestin, Yann Guiguen, Edwige Quillet, Florence Phocas.

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
