## [Decision Letter · Decision Letter 0]

9 Dec 2024

PONE-D-24-47853In-depth investigation of genome to refine QTL positions for spontaneous sex-reversal in XX rainbow troutPLOS ONE

Dear Dr. Phocas,

Thank you for submitting your manuscript to PLOS ONE. After careful consideration, we feel that it has merit but does not fully meet PLOS ONE’s publication criteria as it currently stands. Therefore, we invite you to submit a revised version of the manuscript that addresses the points raised during the review process.

The manuscripts represents high quality work of considerable interest to the readers of PlosOne. The concerns of the reviewers and editors center on flow and writing, not design or analyses. The main issues are the following.

Provide details on bioinformatic analyses and check on comments about analyses by reviewer 1.Provide better flow (thread) through the manuscript.Rev 2 asks for a broadened discussion of the spontaneous masculinization, in ecological context (can shift balance in discussion towards this point).Related to that (rev2) the interaction of rearing temp and genetic factors, is interesting beyond students of aquaculture. Feel free to integrate these ideas into the discussion, again by removing or condensing other points in that section.The text needs to be shortened (rev 3), but some things also need clarifying (rev 1, mainly to do with analyses and validation of findings).

We look forward to receiving your revised manuscript.

Kind regards,

Arnar Palsson, Ph.D.

Academic Editor

PLOS ONE

“The European Maritime and Fisheries Fund and the French National Government supported the NeoBio project (R FEA470016FA1000008).”

“The European Maritime and Fisheries Fund and the French National Government supported this work (R FEA470016FA1000008). The authors are grateful to the companies “Les Fils de Charles Murgat”, “Aqualande” “Bretagne Truite”, “Font Rome” and “Viviers de Sarrance” for fish samples and data collection.”

“The European Maritime and Fisheries Fund and the French National Government supported the NeoBio project (R FEA470016FA1000008).”

Additional Editor Comments:

The manuscripts represents high quality work of considerable interest to the readers of PlosOne. The concerns of the reviewers and editors center on flow and writing, not design or analyses. The main issues are the following.

1. Provide details on bioinformatic analyses and check on comments about analyses by reviewer 1.

2. Provide better flow (thread) through the manuscript.

3. Rev 2 asks for a broadened discussion of the spontaneous masculinization, in ecological context (can shift balance in discussion towards this point).

4. Related to that (rev2) the interaction of rearing temp and genetic factors, is interesting beyond students of aquaculture. Feel free to integrate these ideas into the discussion, again by removing or condensing other points in that section.

5. The text needs to be shortened (rev 3), but some things also need clarifying (rev 1, mainly to do with analyses and validation of findings).

6. Note, thank reviewers (one named)

Reviewers' comments:

Reviewer's Responses to Questions

**Comments to the Author**

1. Is the manuscript technically sound, and do the data support the conclusions?

Reviewer #1: Yes

Reviewer #2: Yes

Reviewer #3: Yes

2. Has the statistical analysis been performed appropriately and rigorously? 

Reviewer #1: Yes

Reviewer #2: Yes

Reviewer #3: Yes

3. Have the authors made all data underlying the findings in their manuscript fully available?

Reviewer #1: Yes

Reviewer #2: Yes

Reviewer #3: Yes

4. Is the manuscript presented in an intelligible fashion and written in standard English?

Reviewer #1: Yes

Reviewer #2: Yes

Reviewer #3: Yes

5. Review Comments to the Author

Reviewer #1: General comments

The present manuscript aims to replicate and refine the position of QTL previously identified for being associated with sex reversion in rainbow trout. To achieve this goal, the authors deploy a panel of methods that could be summarized in the following sequence: 1-sequencing females with progeny showing extreme (high and low) frequencies of reversed males. 2- Identifying SNPs within QTL regions and polymorphic in the females from (1). 3-Test QTL (Fisher exact test) on six population using SNPs from (2). 4- test all SNPs from (1) withing QTL for association with sex reversion using random forest and DAPC. 5- annotation of genes near candidate SNPs.

The manuscript is well written and quite didactic in the description of the background and methods. However, some important details are still missing in the method section and should be clarified. The methods employed in the manuscript are many, and as reader, I missed a logical thread thought the methods and results section. This could be achieved with a short introduction paragraph at the beginning of each section letting the reader know what type of information is expected from the analysis, and how it complements the analysis from the former section.

The word “validation” is used along the manuscript to describe different things, this has a potential to confuse the reader. There is a general aim to validate the QTLs from Fraslin et al. There is also an internal validation dataset within random forest. And finally, from my understanding you used a separate discovery and validation population for the Fisher test, however this remains to be clarified. I found it difficult to grab how the loci detected in the discovery population were matched in the validation group. The only mention of it is in Line 219, but I found it unclear.

Specific comments

Line 165-176. Here you skip many details about the sequence mapping, SNP calling and SNP filtering methods. I think that the exact parameters used in GATK, SAMtools etc should be given in the method section. In addition, the number of SNPs/indel reported here seems very high (about one for every 80bp if we consider the USDA_OmykA_1.1 genome is 2.3Gb long). This number is not completely surprising for a brut output from Mpileup, however the 21.10^6 SNP remaining after filtering (still one for every 110 bp), sounds like a very high density of good quality SNPs. I would therefore like to have a description of the parameters/ criterion that were used for filtering the SNPs as well as some basic statistics from the mapping (mean coverage ±sd).

Regarding the analysis performed with the SNPs from full sequence data (random forest & DAPC), you only use the genomic regions reported previously by Fraslin et al, however, if I understand correctly, you have sequenced the full genome of the females. You could thus compare the distribution of the test between putative QTL region and non-QTL regions. This would provide with an empirical null distribution of the test (Gini index & allele contribution to discriminant axes).

Table3: I don’t understand the reason for splitting the Omy1 QTL into two where Omy1-a stops at 67,960,401 bp and Omy1-b starts at the next base 67,960,402. I also see that each QTL has one given position (based on the Arlee reference genome), but despite the table legend, I didn’t find any comparison between these QTL positions and the QTL reported by Fraslin et al.

Line 384: This could be reformulated. How did the present results make LOC110527930 a less good candidate and tlx1, fbxw4… better ones? Fig3 is a dapc plot, I am not sure how it illustrates the point. (Figure numbers might have been mixed in the text, please check)

Line 419 ACP PCA

Line 419: What are “all those variants”? Is that the set of 33,731 SNPs? You use different subsets of data and analysis methods along the manuscript. It is thus a good idea to keep using precise wording and not hesitate to remind the reader which data you are currently commenting.

Reviewer #2: The article by Dehaullon and collaborators investigates the mechanisms underlying spontaneous masculinization in XX all-female populations of rainbow trout (Oncorhynchus mykiss). Despite the species having an XX/XY sex determination system, a small proportion of males or intersex individuals are consistently observed in these populations. This phenomenon, attributed to heritable minor sex-modifier genes, poses challenges for selective breeding and aquaculture practices. The authors focused on validating previously identified DNA markers associated with spontaneous masculinization in six French farmed populations, targeting QTL regions on chromosomes Omy1, Omy12, and Omy20.

The genome-based approaches and statistical analyses used by the authors have effectively identified functional candidate genes that may influence the reduction of germ cell proliferation and the repression of oogenesis in XX individuals. The methodology used is particularly novel, and the large volume of data generated is a significant strength of the study. Genes such as syndig1, tlx1, and hells on Omy1, along with khdrbs2 and csmd1 on Omy20, have been highlighted for further functional studies, including their interaction with environmental factors like rearing temperature. These findings provide valuable insights for improving sex-ratio control and producing more consistent all-female populations, which are preferred in aquaculture due to their delayed maturation and reduced susceptibility to diseases.

The work is highly interesting and generates a substantial amount of useful information.

I have a few minor comments for consideration:

• The authors highlight that spontaneous masculinization is a highly heritable trait controlled by minor sex-modifier genes, with several QTL regions previously identified. While the importance of monosex populations in aquaculture is well established, the implications of this phenomenon in wild populations require further exploration. Even if spontaneous masculinization occurs at a low frequency in nature, its potential effects on population dynamics, genetic diversity, or ecological stability are worth investigating. Addressing these aspects would not only broaden the ecological context of the study but also underline the relevance of understanding this process beyond its aquaculture applications.

• The conclusion highlights the challenge of undesirable masculinized progeny and the need to better understand the combined effects of genetic and environmental factors on spontaneous maleness in all-female stocks. Given the role of temperature in sex reversal, I suggest that future research could explore in more detail the interactions between genetics and rearing temperatures. Specifically, it would be interesting to consider how these factors might be controlled to achieve the desired all-female progeny, particularly when using free-hormone neomales. Additionally, it could be valuable to outline potential experimental approaches to assess the influence of rearing temperatures at different developmental stages. Exploring whether specific temperature thresholds or genetic combinations could minimize masculinization while optimizing production efficiency may also provide important insights. I believe that discussing these potential avenues for future research, based on the findings of this study, would help to further contextualize the results and expand their applicability to sex ratio management in aquaculture.

I trust that addressing these aspects will further strengthen the manuscript and provide additional perspectives on how to advance the field.

Reviewer #3: The proposed manuscript “In-depth investigation of genome to refine QTL positions for spontaneous sex-reversal in XX rainbow trout” significantly contributes to the study of evolutionary biology, particularly in understanding the evolution of sex determination and its pathways. The manuscript is technically sound and written in high-quality English. Impact of the study is well highlighted and the statistical analyses are appropriately chosen and used. It would be beneficial if the authors included Sanger sequencing results for selected SNPs, especially those that are 100% sex-linked. However, the manuscript is undoubtedly of high quality and meets the requirements for the Plos One journal. I believe it will also be of interest to readers of the journal. I have only minor suggestions:

1) Manuscript is very long and requires the readers’ full intention. The introduction spans more than three pages, and the discussion is more than 9 pages long. I think this extensive length may diminish the reader's interest. Some parts are explained in great detail and I suggest shortening these sections where possible.

2) Line 71: “primary sex determination”. Primary sexual differentiation sounds better to me.

3) Line 100: “follows” instead of “follow”?

4) Line 101: “testis or ovary” instead of “testis and ovary”?

5) Some abbreviations are defined more times (e.g., GSD). Some abbreviations do not have definitions (TGF-β, QTL).

6) Line 170: A period is missing behind a citation.

7) Line 612-613: If I follow this sentence well, it says that the gene gld-1 is localized in cytoplasm. It needs to be reformulated.

8) Line 628: “fertile” is duplicated

6. PLOS authors have the option to publish the peer review history of their article (what does this mean? ). If published, this will include your full peer review and any attached files.

**Do you want your identity to be public for this peer review?** For information about this choice, including consent withdrawal, please see our Privacy Policy .

Reviewer #1: No

Reviewer #2: No

Reviewer #3: **Yes: ** Martin Knytl

---

## [Author Response · Author response to Decision Letter 1]

12 Feb 2025

Dear Editor and reviewers,

We would like to thank you and the three reviewers for your positive’s comments on our manuscript.

Please find attached the revised version of our manuscript “In-depth investigation of genome to refine QTL positions for spontaneous sex-reversal in XX rainbow trout”. The manuscript has been revised according to all the suggestions given by the reviewers. We thank the three reviewers for their valuable comments and suggestions that, we believe, considerably improved the clarity of the manuscript.

One of the main comments from reviewers and the editor was the length of the manuscript, especially the discussion and introduction. We did our best to shorten those two sections by removing details, however we believe that there is a lot of interesting points to mention in the discussion that we have kept. We also broadened the discussion by adding a section on the importance of the spontaneous masculinization in ecological context as requested by R#2 and the editor. Thus we have not been able to reduce the discussion as much as we would have wanted. Yet we hope that the effort we made are sufficient and that this manuscript now meets PLOS ONE’s publication criteria and you will consider it for publication.

In addition, for the role of funders, we declare that the funders had no role in our study. Therefore, the statement to be changed on our behalf on the online submission form is: “The European Maritime and Fisheries Fund and the French National Government supported the NeoBio project (R FEA470016FA1000008). The funders had no role in study design, data collection and analysis, decision to publish, or preparation of the manuscript."

Below, we provide the detailed responses to the reviewer's comments. We started with the main 5 points you, the editor, raised as main issues and have then numbered the reviewers’ comments to facilitate further discussion. All the changes made in the manuscript are displayed as red highlights in the track changes version.

Yours sincerely,

Dr Clémence Fraslin, Audrey Dehaullon and Dr Florence Phocas

The manuscript represents high quality work of considerable interest to the readers of PlosOne. The concerns of the reviewers and editors centre on flow and writing, not design or analyses. The main issues are the following.

1. Provide details on bioinformatic analyses and check on comments about analyses by reviewer 1.

We added details on the bioinformatics analyses, mainly the variant calling and quality controls. The detailed answer to Reviewer 1 is available under their point 3.

2. Provide better flow (thread) through the manuscript.

We have now revised the introduction and the discussion to shorten it, hoping it will improve the flow of the manuscript. We also tried to better explain what we intend to do in the M&M section by adding a few points in the introduction of each sub-section as suggested by R1 as well as adding a new Figure 1 to better follow the process of the different analyses.

3. Rev 2 asks for a broadened discussion of the spontaneous masculinization, in ecological context (can shift balance in discussion towards this point).

4. Related to that (rev2) the interaction of rearing temp and genetic factors, is interesting beyond students of aquaculture. Feel free to integrate these ideas into the discussion, again by removing or condensing other points in that section.

We added a paragraph on this point (masculinisation in an ecological context and impact of the rearing temperature in both farm and ecological context) the end of the discussion section before a shorten conclusion. We added 4 new references to support that discussion. The details of the new paragraph are given in response to reviewer 2 point 2. Since little is known of spontaneous masculinisation of wild rainbow trout and the impact of high temperature on masculinisation is still being investigated, we did not change the focus of the discussion towards this point.

We added four new references to support our discussion.

100. Valdivieso, A., Wilson, C. A., Amores, A., da Silva Rodrigues, M., Nóbrega, R. H., Ribas, L., et al. Environmentally-induced sex reversal in fish with chromosomal vs. polygenic sex determination. Environmental research. 2022 Oct; 213, 113549. doi:10.1016/j.envres.2022.113549

101. Cavileer TD, Hunter SS, Olsen J, Wenburg J, Nagler JJA. Sex-Determining Gene (sdY) assay shows discordance between phenotypic and genotypic sex in wild populations of chinook salmon. Transactions of the American Fisheries Society. 2015 March; 144:423-430. doi: 10.1080/00028487.2014.993479

102. Bezault, E., Clota, F., Derivaz, M., Chevassus, B., & Baroiller, J. F. Sex determination and temperature-induced sex differentiation in three natural populations of Nile tilapia (Oreochromis niloticus) adapted to extreme temperature conditions. Aquaculture. 2007; 272, S3-S16. doi: 10.1016/j.aquaculture.2007.07.227

103. Shinomiya, A., Otake, H., Hamaguchi, S., & Sakaizumi, M. Inherited XX sex reversal originating from wild medaka populations. Heredity. 2010 April; 105(5), 443-448. doi: 10.1038/hdy.2010.51

5. The text needs to be shortened (rev 3), but some things also need clarifying (rev 1, mainly to do with analyses and validation of findings).

We added a figure to better follow the flow of the analyses. We renamed the sections in the methods and added a short introduction paragraph at the beginning of each methods following reviewer 1 recommendation and we believed that this improved the clarity of the manuscript.

We also tried to reduce the introduction and the discussion, hoping to shorten the manuscript. In the introduction, we only succeeded to gain 2 lines as we had to add the full names of genes as recommended by reviewer 3. As we added a new paragraph in the discussion (point 3 and 4 above), the number of lines for “Discussion & conclusion” went in fact from 306 to 307. As we also added details of the variant calling methods (point 1), the efforts made to reduce the initial contents of the introduction and discussion had thus no impact on the length of the manuscript. We are sorry for that, but feel that all the points that are introduced or discussed in the text are needed to help with the understanding of spontaneous masculinization.

We are addressing each reviewers’ comment in more details in the sections below. We included lines numbers from the tracked changes manuscript and reported in the response the additional text that is now included in the manuscript.

Reviewer #1: We thank Reviewer #1 for their appreciation of our manuscript. We think that the reviewer’s comments helped improve the clarity of the manuscript. We addressed R#1 first two main comments and other specific comments below.

1. “The methods employed in the manuscript are many, and as reader, I missed a logical thread thought the methods and results section. This could be achieved with a short introduction paragraph at the beginning of each section letting the reader know what type of information is expected from the analysis, and how it complements the analysis from the former section.”

We thank the reviewer for this useful comment, we have now added sentences in the methods to explain why the approach was used and we also added a graphical summary of the approaches and dataset used in a new Fig 1. We hope that this helped improve the clarity and the flow of the manuscript.

“Fig. 1. Graphical summary of the different statistical analyses performed in the discovery or validation populations.” (Lines 219-220)

“We sought to validate the existence of the four QTL detected by Fraslin et al. [27] as linked to spontaneous sex-reversal of XX trout in six diverse French populations of rainbow trout (Table 1, Fig. 1). Population A was composed of XX sibs from the same birth cohort of the parents used to produce the initial QTL discovery population [27]. Fish from populations B and C and those from populations E and F came from two other French breeding compagnies, while fish from population D came from a commercial site using fry from unknown origin.” (Lines 198-293)

“Exact Fisher test on all SNPs of QTL in 23 extreme dams of the discovery population

In order to refine the QTLs boundaries and confront the results between the validation and discovery populations (Fig. 1), an exact Fisher test was run to detect the association of all the sequence variants observed in the four extended QTL regions with the average progeny sex-ratios of the 23 extreme dams of the discovery population. ” (Lines 234-238)

“To refine the QTL locations in the discovery population using the latest genome assembly we i) used Random Forests approaches with haplotypes and ii) applied a Discriminant Analysis of Principal Components on the sequence data from the 23 dams selected above.” (Lines 247-249)

2. “The word “validation” is used along the manuscript to describe different things, this has a potential to confuse the reader. There is a general aim to validate the QTLs from Fraslin et al. There is also an internal validation dataset within random forest. And finally, from my understanding you used a separate discovery and validation population for the Fisher test, however this remains to be clarified. I found it difficult to grab how the loci detected in the discovery population were matched in the validation group. The only mention of it is in Line 219, but I found it unclear.”

Indeed, we used “validation” to refer to many things in the original manuscript. We made an effort to modify the text to make it clearer.

We are now using “validation” population and “validate” the QTL only when we talk about the “six validation populations” that are distinct from the original discovery population from Fraslin et al [27]. And we use “refine” or “confirm” to describe the new QTL position in the discovery population after mapping to the new genome reference (Arlee). We hope that the new Figure 1 will help clarify that point as well.

Regarding the reviewer’s comment on the validation of loci detected in the discovery population: we rephrased the methods used for the genotyping of the validation population using 96 SNPs arrays containing SNPs identified in the discovery population by Fraslin et al [27].

“Two 96 SNPs genotyping arrays using microfluidic real-time PCR Fluidigm Kasp chemistry were designed. A set of 192 SNPs identified in Fraslin et al [27] (see S1 Supplementary Tables, S2 Table) were selected, corresponding to: 140 SNPs in the two main QTL identified on chromosome Omy1, 19 SNPs in a putative QTL region detected on Omy12 and 33 SNPs in a large 6 Mb region with putative QTL on Omy20.

Pieces of caudal fin sampled from 315 fish, corresponding to at least 30 XX fish per population, including a minimum of eight neomales (Table 1), were sent to Gentyane genotyping platform (INRAE, Clermont-Ferrand, France) for genotyping after DNA extraction using the DNA advance kit from Beckman Coulter following manufacturer instructions.

After quality control, 19 SNPs were eliminated (including 14 from Omy1) from the analyses of all the populations because their genotyping rate was below 90%, 173 SNPs were kept for validation of the QTL.” (Line 198-215)

The methods to validate the relevance of those SNPs using the exact Fisher test are described lines 222 to 244. We added a new header to the section in the methods where we describe another exact Fisher test performed, this time, in the discovery population:” Exact Fisher test on all SNPs of QTL in 23 extreme dams of the discovery population”. We think that the section below is the one the reviewer refers to (previously lines 218-225), we modified this section to make it clearer what test is performed in the discovery population.

Specific comments

3. Line 165-176. Here you skip many details about the sequence mapping, SNP calling and SNP filtering methods. I think that the exact parameters used in GATK, SAMtools etc should be given in the method section. In addition, the number of SNPs/indel reported here seems very high (about one for every 80bp if we consider the USDA_OmykA_1.1 genome is 2.3Gb long). This number is not completely surprising for a brut output from Mpileup, however the 21.10^6 SNP remaining after filtering (still one for every 110 bp), sounds like a very high density of good quality SNPs. I would therefore like to have a description of the parameters/ criterion that were used for filtering the SNPs as well as some basic statistics from the mapping (mean coverage ±sd).

In Gao et al (2018) and Bernard et al. (2022) you will see that the rainbow trout genome is extremely polymorph. In Bernard et al. (2022), the average distance between two successive high-quality variants was 60 bp on the Swanson rainbow trout reference genome, consistent with the value of one SNP every 64 bp previously reported by Gao et al. (2018).

Gao, G., Nome, T., Pearse, D. E., Moen, T., Naish, K. A., Thorgaard, G. H., et al. (2018). A New Single Nucleotide Polymorphism Database for Rainbow Trout Generated Through Whole Genome Resequencing. Front. Genet. 9, 147. doi:10.3389/fgene.2018.00147.

Bernard M., Dehaullon A., Gao G., Paul K., Lagarde H., Charles M., Prchal M., Danon J., Jaffrelo L., Poncet C., Patrice P., Haffray P., Quillet E., Dupont-Nivet M., Palti Y., Lallias D., Phocas F. (2022). Development of a High-Density 665 K SNP Array for Rainbow Trout Genome-Wide Genotyping. Frontiers in Genetics, 13, 941340 https://dx.doi.org/10.3389/fgene.2022.941340

We have now added more details on the new mapping as well as for the variant calling and filtering in methods and the section now reads:

“On average, 9.252 107 paired reads (1.456 107 sd) were mapped on the Arlee genome, corresponding to a mean coverage of 12X. While the percentage of properly paired sequence reads over all sequenced pairs was in average 87.3% for the alignment against Swanson’s genome assembly, these statistics went up to 94.1% for the alignment against Arlee’s genome assembly.

After mapping, duplicates were marked using the MarkDuplicate tool, and base quality score where recalibrated using ApplyBQSR tool from GATK 4.2.2.0 [32], then variants were called independently with the three different variants calling tools HaplotypeCaller from GATK, Freebayes 1.3.5 [33] and SAMtools Mpileup 1.11 [34]. Recommended quality filters were used to ignore low-confidence alignments: a minimum mapping quality of 30, a minimum base quality required to consider a base for calling of 10X reads, minimum phred-scaled confidence threshold at which variants should be called of 30. A total of 29,229,949 variants that were obtained by the three different variant calling tools were kept for further quality controls using vcftools 1.15 and keeping indels and SNPs located only on known chromosomes, removing variants located on un-located contigs or mitochondrial chromosome. Further quality filtering on variant coverage were performed using SelectVariant and Variant Filtration tools (GATK) according to the hard-filtering recommendations. Briefly QD < 2.0 (to normalize variant quality), FS > 6.0 (measure of the Phred-scaled probability that there is strand bias at the site), MQ < 40.0 (mapping quality), SOR > 3.0 (Strand Odds Ratio), MQRankSum < -12.5 (u-based z-approximation from the Rank Sum Test for mapping qualities) and ReadPosRankSum < -8.0 (u-based z-approximation from the Rank Sum Test for site position within reads).” (Lines 166 - 186)

4. Regarding the analysis performed with the SNPs from full sequence data (random forest & DAPC), you only use the genomic regions reported previously by Fraslin et al, however, if I understand correctly, you have sequenced the full genome of the females. You could thus compare the distribution of the test between putative QTL region and non-QTL regions. This would provide with an empirical null distribution of the test (Gini index & allele contribution to discriminant axes).

It would have been difficult or impossible to use the random forest and DAPC approaches on the full genome or even the full chromosome (so including QTL and non-QTL region) due to the limited number of individuals (23 sequenced dams with extreme offspring distribution) we could consider in the analysis. As

---

## [Decision Letter · Decision Letter 1]

4 Mar 2025

PONE-D-24-47853R1In-depth investigation of genome to refine QTL positions for spontaneous sex-reversal in XX rainbow troutPLOS ONE

Dear Dr. Phocas,

Thank you for submitting your manuscript to PLOS ONE. After careful consideration, we feel that it has merit but does not fully meet PLOS ONE’s publication criteria as it currently stands. Therefore, we invite you to submit a revised version of the manuscript that addresses the points raised during the review process.

The text is quite verbose and the results often swing from past to present tense. With specks of MS thesis wording, where the sentences become chatty and winding using “looking into …” and “favorite causative gene” In particular shorten the gene talk section, from line 512 to 737. Not all of those points are of nr. 1 priority. And make sure you tone down talk of causative relationships, the associations are weak and effects sizes generally small. This rewriting effort could be led by the principal investigator, to train the student in a more succint writing style.

Minor points (examples of writing fixes, many other similar instances are found in manuscript, urge you to find them and fix). See accompanying.

We look forward to receiving your revised manuscript.

Kind regards,

Arnar Palsson, Ph.D.

Academic Editor

PLOS ONE

Journal Requirements:

Additional Editor Comments:

PLOSsex2

The manuscript is greatly improved and the two reviewers were mostly content with the changes made.

The reviewers are generally happy, except with the length and lack of definitions of abbreviations.

Main point

The text is quite verbose and the results often swing from past to present tense. With specks of MS thesis wording, where the sentences become chatty and winding using “looking into …” and “favorite causative gene” In particular shorten the gene talk section, from line 512 to 737. Not all of those points are of nr. 1 priority. And make sure you tone down talk of causative relationships, the associations are weak and effects sizes generally small. This rewriting effort could be led by the principal investigator, to train the student in a more succint writing style.

Minor points (examples of writing fixes, many other similar instances are found in manuscript, urge you to find them and fix).

Line 32

Is “located” needed?

Line 36

“to precise their functional roles as” strange wording?

Line 59 and 62

Add references. Check if further citations are needed for other factual statements.

Line82

“is changed to the opposing pathway,” can you reword.

Line 120

Turn into past tense “Currently, there are” to “…have been”?

In all animals or just fish?

Line 135

Reword “is a repeatedly phenomenon observed “

Line 163-4

Fix sci. annotation of numbers “9.252 107 “

Line 209

Add space “instructions.After”

Line 214

Spell check.

Line 325

Reword “Among the latter,” and “had at least an effect”

Line 333

Use regular PLOS one style when referrring to supplements “given in S1 Supplementary Tables, S3 Table”

Line 333-35 “with indication of the level of significance in those populations, as well as the corresponding positions of SNPs on both reference genomes” this can be skipped.

Line 335

Use “two or more” rather than “at least two” Check if this wording is used elsewhere!

Line 341

In table 2 there is some ambiguity on the annotation. “Close to …” is used in some cases but in other exact distance (in kb) is used. Use one style or the other.

Line 347

Shorten “for the QTL that are given in Table 3” to “for the QTL (Table 3)” Please see if you can shorten the manuscript with finding more examples of “extra words” of this type.

Line 349

Reword “we enlarged the search of QTL in three different regions”

Line 355

Suggest title “redefined boundaries…” Also, suggest that you add columns with the “old” boundaries, instead of using footnotes. Makes the data more easily apprehended and reused.

Line 367

Spell out Random Forest (RF) here. Also, it is not clear from the text that this is a sliding window approach (can you specify the window size here?)

Line 384

Drop “we estimated”

Line 389

Verbose sentence, can you shorten? “Due to the very small size of our dataset, it is likely that we had too many variables in the full RF analysis, creating some « noise » by considering over 1,100 haplotypes together in the analysis to classify our 23 individuals”

Line 392

Shorten “Looking at the locations (Fig. 2) of the best ten RF ranked haplotypes on the 2 Mb evaluated on Omy1 (see S1 Supplementary Tables, S5 Table), they fell” to

“The best ten RF ranked haplotypes in the 2 Mb on Omy1 (Fig2, see S1 Supplementary Tables, S5 Table), fell”

Line 410

Can drop “All”

Line 413

Reword, don’t think you should have a favorite gene! “initially was our favourite causative gene”

Line 416

Reword “the first three best haplotypes to explain sex-reversal were”

Line 420

Strange wording of sentence, “unfortunately” should not be used!

Figure 2 and 3 legend. The RF colours score is not explained, is red or blue associating??

Figure 2 and 3. Related to above, maybe a unicolour scale makes more sense than a two colour scale, since this denotes a scale from 0-1? Also, the lines corresponding to each panel, Fisher primary, discorvery DAPC are not clearly explained. Why are there two lines for FP, and one for DAPC? Should the different regions be labelled A, B, C? (with OmyID as secondary headings?)

Line 430

Strange wording “of the best haplotypes to explain sex-reversal across all the”

Line 436

Reword “appeared then as the subsequent haplotypes ranked” to “appeared in the RF haplotype analyses, ranked”

Line 438

Strange wording “best-ranked”, find alternative and fix throughout manuscript!

Line 450

Reword “On the contrary, the DAPC allowed us to perfectly discriminate” to “On the contrary, the DAPC perfectly discriminated…” Many more cases where the text can be changed from first person to passive description, and extra words trimmed. Like the sentence preceeding this one.

Line 455

Indicate how many variants, “PCA of the XX variants in the” and replace “best” with some other word.

Line 459

Reword this sentence “In S1 Supplementary Tables, S10 Table are given the genotypes for the 100 best SNPs that …” this is example of “Thesis” writing, that can be shortened and made more consise.

Line 478

Replace “playing a role on” with “associating with” and state that this is “female to male” reversal!

Line 500

Shorten and reword to “was the motivation to further investigate the variants in larger genomic”

Line 505

Reword “population A although it was closely related”

Line 527

Most likely, not very likely.

Line 613

Put table 5 as a supplemental table. Is of minor importance!

Line 620

Reword “has been pointed out as one of”

Line 741

Reword “this genetic influence”

Line 749

Reword to “Indeed, the high heritability of spontaneous maleness in XX individuals [27]”

Line 751

Check verb “Improved” and drop “then” from sentence.

Line 773

Reword to “we found in several French… that variants in some chromosomes associated…”

Reviewers' comments:

Reviewer's Responses to Questions

**Comments to the Author**

1. If the authors have adequately addressed your comments raised in a previous round of review and you feel that this manuscript is now acceptable for publication, you may indicate that here to bypass the “Comments to the Author” section, enter your conflict of interest statement in the “Confidential to Editor” section, and submit your "Accept" recommendation.

Reviewer #1: All comments have been addressed

Reviewer #2: All comments have been addressed

Reviewer #3: All comments have been addressed

2. Is the manuscript technically sound, and do the data support the conclusions?

Reviewer #1: Yes

Reviewer #2: Yes

Reviewer #3: Yes

3. Has the statistical analysis been performed appropriately and rigorously? 

Reviewer #1: Yes

Reviewer #2: Yes

Reviewer #3: Yes

4. Have the authors made all data underlying the findings in their manuscript fully available?

Reviewer #1: Yes

Reviewer #2: Yes

Reviewer #3: Yes

5. Is the manuscript presented in an intelligible fashion and written in standard English?

Reviewer #1: Yes

Reviewer #2: Yes

Reviewer #3: Yes

6. Review Comments to the Author

Reviewer #1: all comments have been addressed and the manuscript was modified accordingly. I have no further request.

Reviewer #2: All suggested changes and recommendations have been incorporated into the revised version of the manuscript. The authors have addressed all reviewer comments, and the manuscript has been improved accordingly.

Reviewer #3: The proposed manuscript has been revised and improved. The authors have addressed all the recommendations and I agree with their responses. While the article remains lengthy, I understand that it is necessary to provide a comprehensive understanding of the multiple topics covered in the manuscript. However, one suggestion still needs to be considered: not all abbreviations have been defined. Please define the abbreviation ‘omy’ in both the abstract and the introduction, and ‘QTL’ in the abstract. I believe that the study has a sufficiently high impact and fits well with the scope of the PLoS One journal. I look forward to seeing the published version of the paper.

7. PLOS authors have the option to publish the peer review history of their article (what does this mean? ). If published, this will include your full peer review and any attached files.

**Do you want your identity to be public for this peer review?** For information about this choice, including consent withdrawal, please see our Privacy Policy .

Reviewer #1: No

Reviewer #2: **Yes: ** Alejandro S. Mechaly

Reviewer #3: **Yes: ** Martin Knytl

---

## [Author Response · Author response to Decision Letter 2]

30 Mar 2025

Dear Editor,

We thank you and the reviewers for your assessment of our revised manuscript. We have implemented the changes required by the editor and have further edited the text to i) simplify sentences, ii) use passive form and iii) further reduce the discussion. We hope you find the new version improved and suitable for publication.

The changes that were made are detailed below in our answer to your comments and we provided a tracked changes version of the manuscript.

We would like to point out that, contrary to your comment no student was involved in writing the manuscript. Dr Florence Phocas, the principal investigator wrote the original draft with the help of Dr Clémence Fraslin, a senior post-doctoral fellow. And that both Dr Phocas and Dr Fraslin were responsible for the correction made after review. We understand that, as none of us are native speaker, the language needed some editing and we hope that the new version provided is up to the journal standards.

Finally, we would like to say that we disagree with your statement “the associations are weak and effects sizes generally small”. When you find associations across several populations and small data sets, it means that the associations are not weak and even more consistent. Thus, we did not really tone down the discussion on “causative relationships” as we believe that already we were cautious in our approach using phrases as “we hypothesized”

Best regards,

Dr Florence Phocas and Dr Clémence Fraslin

Additional Editor Comments:

PLOSsex2

The manuscript is greatly improved and the two reviewers were mostly content with the changes made.

We thank you and the reviewers for the appreciation of the quality of the revised version of the manuscript.

The reviewers are generally happy, except with the length and lack of definitions of abbreviations.

We added the requested definition for the abbreviation of QTL in the abstract. Regarding the use of “OmyXX” for chromosome numbers, this is the standard name for rainbow trout chromosomes. We added the scientific name “Oncorhynchus mykiss” in the abstract and hope this is enough to clarify where “Omy” is derived from.

We made further cuts in the discussion and in the manuscript to reduce the length of the new revised version.

Main point

The text is quite verbose and the results often swing from past to present tense. With specks of MS thesis wording, where the sentences become chatty and winding using “looking into …” and “favorite causative gene” In particular shorten the gene talk section, from line 512 to 737. Not all of those points are of nr. 1 priority. And make sure you tone down talk of causative relationships, the associations are weak and effects sizes generally small. This rewriting effort could be led by the principal investigator, to train the student in a more succint writing style.

We made the suggested comments (see details below) and we also carefully revised the manuscript and modified the text to reduce the word count as suggested by the editor. We reduced the discussion by deleting sentences and we removed a discussion paragraph (lines 658 to 666).

Minor points (examples of writing fixes, many other similar instances are found in manuscript, urge you to find them and fix).

We made all the correction required by the editor (details below) and we also made more correction (fixing the tense, the sentences and reducing the word count) throughout the manuscript. Those modifications are highlighted with tracked changes in the manuscript.

Line 32

Is “located” needed?

Indeed, it’s not needed and probably comes from editing the text and removing “within the QTLs”. We have now removed “located”.

Line 36

“to precise their functional roles as” strange wording?

Indeed, the sentence has been modified and now reads: “further investigation to validate their potential sex-modifier roles as well as their interaction with rearing temperature.”

Line 59 and 62

Add references. Check if further citations are needed for other factual statements.

We added three new references here [4,5,6] (line 60) and reported the reference [2] line 63.

4. Aksnes A, Gjerde B, Roald S O. Biological, chemical and organoleptic changes during maturation of farmed Atlantic salmon, Salmo salar. 1986. Aquaculture, 53(1), 7-20. doi: 10.1016/0044-8486(86)90295-4

5. Cleveland B M, Kenney P B, Manor M L, Weber G M. Effects of feeding level and sexual maturation on carcass and fillet characteristics and indices of protein degradation in rainbow trout (Oncorhynchus mykiss). 2012. Aquaculture, 338, 228-236. doi: 10.1016/j.aquaculture.2012.01.032

6. Manor M L, Weber G M, Salem M, Yao J, Aussanasuwannakul A, Kenney P B. Effect of sexual maturation and triploidy on chemical composition and fatty acid content of energy stores in female rainbow trout, Oncorhynchus mykiss. 2012. Aquaculture, 364, 312-321. doi: 10.1016/j.aquaculture.2012.08.012

Line82

“is changed to the opposing pathway,” can you reword.

The sentence has been reworded to “Spontaneous sex reversal occurs because of a failure to maintain the initiated pathway or a failure to repress the opposite pathway [6], switching the sexual phenotype of the organism to the opposite sex.”

Line 120

Turn into past tense “Currently, there are” to “…have been”?

In all animals or just fish?

Changed to “have been” and sentence added “discovered in fish, mammals, birds and frogs”.

Line 135

Reword “is a repeatedly phenomenon observed “

Reworded: “phenomenon that has been repeatedly observed”

Line 163-4

Fix sci. annotation of numbers “9.252 107 “

Scientific annotation of numbers has been fixed: “9.252E+7” an “1.456E+7”

Line 209

Add space “instructions.After”

Space has been added

Line 214

Spell check.

The spelling of “company” was changed to “company”.

Line 325

Reword “Among the latter,” and “had at least an effect”

Reworded: “117 had a significant effect in at least one population with 50 SNPs on Omy1 having an effect in population A, that was genetically close to the QTL discovery population.”

Line 333

Use regular PLOS one style when referrring to supplements “given in S1 Supplementary Tables, S3 Table”

Modified to “Table C in S1 Tables” here and throughout the manuscript. All Supplementary tables were renamed to match this new style as per PLOS one recommendations to authors.

Line 333-35 “with indication of the level of significance in those populations, as well as the corresponding positions of SNPs on both reference genomes” this can be skipped.

The sentence has now been removed.

Line 335

Use “two or more” rather than “at least two” Check if this wording is used elsewhere!

“at least two” has been changed to “two or more” lines 335, 341, and 462 (“at least six offspring” changed to “six offspring or more”) and line 480 and is not used elsewhere in the manuscript.

Line 341

In table 2 there is some ambiguity on the annotation. “Close to …” is used in some cases but in other exact distance (in kb) is used. Use one style or the other.

The style has been harmonized in the table 2.

Line 347

Shorten “for the QTL that are given in Table 3” to “for the QTL (Table 3)” Please see if you can shorten the manuscript with finding more examples of “extra words” of this type.

The sentence has been changed to the proposed version and similar changes were made throughout the manuscript.

Line 349

Reword “we enlarged the search of QTL in three different regions”

The sentence has been modified “QTL in three different regions of Omy20, were enlarged, spanning”

Line 355

Suggest title “redefined boundaries…” Also, suggest that you add columns with the “old” boundaries, instead of using footnotes. Makes the data more easily apprehended and reused.

Table 3 was modified: the title was changed according to the editor’s comment and a new column “Old boundaries on Swanson reference” was added. With a new legend: “Swanson reference assembly genome: Omyk_1.0 (GenBank assembly accession GCA_002163495.1)”

Line 367

Spell out Random Forest (RF) here. Also, it is not clear from the text that this is a sliding window approach (can you specify the window size here?)

Random Forrest has been spelled out. The window are fixed windows of 80 SNPs and the explanation can be found lines 282-284 “The final parameter set (swind=40, ntree=400 and mtry=7) was chosen as consistently giving the lower median error rate over 50 runs for any of the three tested chromosomes.”

Line 384

Drop “we estimated”

“we estimated” has been removed

Line 389

Verbose sentence, can you shorten? “Due to the very small size of our dataset, it is likely that we had too many variables in the full RF analysis, creating some « noise » by considering over 1,100 haplotypes together in the analysis to classify our 23 individuals”

Rephrased: “Due to the very small size of our dataset (n=23), it is likely that having over 1,100 haplotypes to consider in the full RF analysis created some “noise” in the classification process.”

Line 392

Shorten “Looking at the locations (Fig. 2) of the best ten RF ranked haplotypes on the 2 Mb evaluated on Omy1 (see S1 Supplementary Tables, S5 Table), they fell” to

“The best ten RF ranked haplotypes in the 2 Mb on Omy1 (Fig2, see S1 Supplementary Tables, S5 Table), fell”

We rephrased the sentence that now reads: “The best ten RF ranked haplotypes in the 2 Mb evaluated on Omy1 (see S1 Supplementary Tables, S5 Table) fell within the nine following genes (Fig.2): sdhaf4, tlx1, syndig1 and vsx1 genes located in QTL Omy1_a; cep68, fbxw4, hells, LOC110527930, and gbf1 in QTL Omy1_b. Six of those genes have been identified with significant effects on sex-reversal in several validation populations (Table 2) or within some significant SNPs in the discovery population (Table 4).”

Line 410

Can drop “All”

“All” was removed

Line 413

Reword, don’t think you should have a favorite gene! “initially was our favourite causative gene”

This has been changed in “the best candidate gene”.

Line 416

Reword “the first three best haplotypes to explain sex-reversal were”

Rephrased as: “the first three haplotypes most likely explain sex-reversal”

Line 420

Strange wording of sentence, “unfortunately” should not be used!

The paragraph has been entirely rephrased and now reads: “On Omy20 (Fig.3) the best ten ranked haplotypes in a 5Mb window investigated in the RF analysis (S7 Table) were overlapping with the following eight genes, presented in decreasing order of importance: lrrc59, csmd1, khdrbs2, abcg5, akt3, rgs17, caskin2 and dst. No variants from this region could be tested in the validation populations, however, some highly significant SNPs are detected in the discovery population for four of these genes (akt3, khdrbs2, dst and csmd1, see Table 4).”

Figure 2 and 3 legend. The RF colours score is not explained, is red or blue associating??

Figure 2 and 3. Related to above, maybe a unicolour scale makes more sense than a two colour scale, since this denotes a scale from 0-1? Also, the lines corresponding to each panel, Fisher primary, discorvery DAPC are not clearly explained. Why are there two lines for FP, and one for DAPC? Should the different regions be labelled A, B, C? (with OmyID as secondary headings?)

For Figures 2 and 3, the colours are “dark red = highest ranked haplotype through RF, to dark blue: lowest ranked haplotype”. A Legend has been added to explain the figures. We do not think a unicolour scale would be wise as there is up to 17 different RF to rank for a given chromosome and such long unicolor scale are difficult to see.

Line 430

Strange wording “of the best haplotypes to explain sex-reversal across all the”

The sentence was rephrased: “When all the haplotypes were analyzed together across all the QTL regions spanning the 3 chromosomes (S8 Table) and ranked through RF […]”

Line 436

Reword “appeared then as the subsequent haplotypes ranked” to “appeared in the RF haplotype analyses, ranked”

The sentence was modified following your suggestion.

Line 438

Strange wording “best-ranked”, find alternative and fix throughout manuscript!

“best-ranked” was replaced by “top-ranked” throughout the manuscript. And the sentence was rephrased as follow “Noticeably, most of the top-ranked haplotypes involved sequences located within genes, and the ranking was consistent between the full RF and the RF limited to their chromosomic location”.

Line 450

Reword “On the contrary, the DAPC allowed us to perfectly discriminate” to “On the contrary, the DAPC perfectly discriminated…” Many more cases where the text can be changed from first person to passive description, and extra words trimmed. Like the sentence preceeding this one.

This sentence has been changed and other changes to passive description were made throughout the manuscript.

Line 455

Indicate how many variants, “PCA of the XX variants in the” and replace “best” with some other word.

the number of variants (27,828) has been added and “best” is now replace by “top” here and throughout the manuscript.

Line 459

Reword this sentence “In S1 Supplementary Tables, S10 Table are given the genotypes for the 100 best SNPs that …” this is example of “Thesis” writing, that can be shortened and made more consise.

The new sentence reads: “The genotypes for the 100 top SNPs (from 27,828, SNPs) that were genotyped for the 23 dams, as well as for five additional dams, are given in Table S10. These five dams, with at least six offspring with a record for phenotypic sex the discovery population (see S1 Table), were used as an internal validation : three dams (noted AS, AT and AU) with high progeny sex-reversal ratio and two dams (noted BS and BT) with no sex-reversed offspring.”

Line 478

Replace “playing a role on” with “associating with” and state that this is “female to male” reversal!

The sentence was rephrased and now reads: “All the QTL regions previously identified [27] were confirmed as associated with female t0 male sex-reversal in two or more populations different from the discovery one.”

Line 500

Shorten and reword to “was the motivation to further investigate the variants in larger genomic”

Rephrased as “were the motivations to enlarge the genomic region investigated in the current study.”

Line 505

Reword “population A although it was closely related”

The sentence has been rephrased

Line 527

Most likely, not very likely.

this has been modified

Line 613

Put table 5 as a supplemental table. Is of minor importance!

Table 5 has now been changed as Table K in S1 Tables and removed from the main text.

Line 620

Reword “has been pointed out as one of”

rephrased as “identified”

Line 741

Reword “this genetic influence”

Reworded as “the genetic influence”

Line 749

Reword to “Indeed, the high heritability of spontaneous maleness in XX individuals [27]”

Reworded as suggested by the editor

Line 751

Check verb “Improved” and drop “then” from sentence.

“Improved” has been corrected and “then” removed from the sentence

Line 773

Reword to “we found in several French… that variants in some chromosomes associated…”

The sentence has been rephrased: “We confirmed on several French commercial rainbow trout populations the importance of some genomic regions on chromosomes Omy1, Omy12 and Omy20 for spontaneous masculinization of XX-individuals.Those regions can be considered as minor sex-determining regions in rainbow trout.”

6. Review Comments to the Author

Reviewer #1: all comments have been addressed and the manuscript was modified accordingly. I have no further request.

Reviewer #2: All suggested changes and recommendations have been incorporated into the revised version of the manuscript. The authors have addressed all reviewer comments, and the manuscript has been improved accordingly.

Reviewer #3: The proposed manuscript has been revised and improved. The authors have addressed all the recommendations and I agree with their responses. While the article remains lengthy, I understand that it is necessary to provide a comprehensive understanding of the multiple topics covered in the manuscript. However, one suggestion still needs to be considered: not all abbreviations have been defined. Please

---

## [Editor Report · Decision Letter 2]

1 Apr 2025

In-depth investigation of genome to refine QTL positions for spontaneous sex-reversal in XX rainbow trout

PONE-D-24-47853R2

Dear Dr. Phocas,

We’re pleased to inform you that your manuscript has been judged scientifically suitable for publication and will be formally accepted for publication once it meets all outstanding technical requirements.

Kind regards,

Arnar Palsson, Ph.D.

Academic Editor

PLOS ONE
---

## [Editor Report · Acceptance letter]

PONE-D-24-47853R2

PLOS ONE

Dear Dr. Phocas,

I'm pleased to inform you that your manuscript has been deemed suitable for publication in PLOS ONE. Congratulations! Your manuscript is now being handed over to our production team.

Kind regards,

on behalf of

Dr. Arnar Palsson

Academic Editor

PLOS ONE